# ECE2 regulates neurogenesis and neuronal migration during human cortical development

Isabel Y Buchsbaum[1,2] (ID), Pavel Kielkowski[3], Grazia Giorgio[1,2], Adam C O'Neill[4], Rossella Di Giaimo[1,5], Christina Kyrousi[1], Shahryar Khattak[6], Stephan A Sieber[3], Stephen P Robertson[4] & Silvia Cappello[1,*] (ID)

## Abstract

During embryonic development, excitatory projection neurons migrate in the cerebral cortex giving rise to organised layers. Periventricular heterotopia (PH) is a group of aetiologically heterogeneous disorders in which a subpopulation of newborn projection neurons fails to initiate their radial migration to the cortex, ultimately resulting in bands or nodules of grey matter lining the lateral ventricles. Although a number of genes have been implicated in its cause, currently they only satisfactorily explain the pathogenesis of the condition for 50% of patients. Novel gene discovery is complicated by the extreme genetic heterogeneity recently described to underlie its cause. Here, we study the neurodevelopmental role of *endothelin-converting enzyme-2* (*ECE2*) for which two biallelic variants have been identified in two separate patients with PH. Our results show that manipulation of *ECE2* levels in human cerebral organoids and in the developing mouse cortex leads to ectopic localisation of neural progenitors and neurons. We uncover the role of *ECE2* in neurogenesis, and mechanistically, we identify its involvement in the generation and secretion of extracellular matrix proteins in addition to cytoskeleton and adhesion.

**Keywords** cerebral organoids; endothelin-converting enzyme-2; human cortical development; neuronal migration disorders; periventricular heterotopia
**Subject Categories** Development; Neuroscience

## Introduction

Mammalian neocortical development occurs through a highly complex series of processes including neural progenitor proliferation, migration of newborn neurons towards the future cortical plate (CP), and their terminal maturation and integration into neuronal networks [1,2]. Neuroepithelial cells give rise to apical radial glia (aRG), bipolar neural stem cells of the ventricular zone (VZ) spanning the thickness of the developing cortex, which can self-renew to increase the size of the neural progenitor cell (NPC) pool and differentiate to produce basal progenitors (BPs)—including intermediate progenitor cells (IPCs) and basal radial glia (bRG)—or neurons directly [1,3,4]. Newly generated projection neurons acquire a bipolar morphology and use the basal processes of RG as a scaffold for their radial migration towards the developing CP [5].

The widespread occurrence of malformations during cortical development underscores the importance of this tightly controlled process of events. One of the most common neuronal migration disorders (NMDs) is periventricular nodular heterotopia (PH) [6]. PH is characterised by a subset of cortical neurons failing to initiate radial migration towards the CP and instead remaining as nodules at the apical, ventricular surface where they were generated. Patients with PH can present with intellectual disability and epilepsy [3]. Bands or nodules of grey matter in contact with the normotopic overlaying or further distant cortex may represent the source of epileptogenic activity in patient brains, although clinical presentations are highly heterogeneous [7,8].

Known molecular causes of the failure of neurons to migrate include an intrinsic disruption in the cytoskeleton of neurons (cell-autonomous mechanism) or a disruption in the RG scaffold that they use for their radial migration (non-cell-autonomous mechanism) [9–13]. Until recently, variants in only a handful of genes have been shown to cause PH. These include X-linked *FLNA* [14–17] as the most frequent cause, rare biallelic variants in *ARFGEF2* [15,18–22], *MCPH1* [23], *FAT4* [10,13,24], *DCHS1* [10,13] and *INTS8* [25] as cause of recessive forms of PH. *De novo* or inherited variants in *NEDD4L* [26] or *MAP1B* [27] also cause heterozygous forms of this disorder. Recently, high-throughput analysis of patients with PH has revealed extreme genetic heterogeneity including single nucleotide and copy-number variants [27–30]. Functional analyses have delivered supportive evidence for the potential for further candidate genes to be causative, with some of them implicating a role for bRG cell dysfunction in its pathogenesis [29].

1   Max Planck Institute of Psychiatry, Munich, Germany
2   Graduate School of Systemic Neurosciences, Ludwig-Maximilians-University Munich, Planegg, Germany
3   Department of Chemistry, Chair of Organic Chemistry II, Center for Integrated Protein Science (CIPSM), Technische Universität München, Garching bei München, Germany
4   Department of Women's and Children's Health, University of Otago, Dunedin, New Zealand
5   Department of Biology, University of Naples Federico II, Naples, Italy
6   DFG Center for Regenerative Therapies, Dresden (CRTD), Technische Universität Dresden, Dresden, Germany
    *Corresponding author. Tel: +49 (0) 89-30622-253; E-mail: silvia_cappello@psych.mpg.de

In this study, we combined human-induced pluripotent stem cell (iPSC)-based 2D and 3D *in vitro* models [31,32] with acute *in vivo* manipulation of gene expression in the developing mouse cortex [33] to characterise the role of endothelin-converting enzyme-2 (*ECE2*) in cortical development, for which two patients with PH have been identified with rare biallelic variants [27]. The applicability of cerebral organoids (COs) [32] to model different aspects of PH and explore the role of novel candidate causative genes in human cortical development has been shown in several recent studies [13,28,29]. We identified a role for *ECE2* both in RG polarity and morphology, and in neuronal migration and differentiation. Proteome analysis of genome-engineered *ECE2* knockout (KO) iPSC-derived COs revealed an additional function in the production and secretion of components of the extracellular matrix (ECM) and additionally suggested an involvement of ECE2 in protein phosphorylation.

## Results

### Identification of *ECE2* as candidate gene for periventricular heterotopia

Recent trio-based exome sequencing studies have indicated considerable genetic heterogeneity underlying PH, complicating the identification of causative genes given the large mutational landscape [27]. To aid novel disease gene identification, we sought to functionally complement such large-scale sequencing efforts with evidence obtained from both *in vivo* and *in vitro* model systems. Using data obtained from a recent study in which 202 individuals with PH were subject to whole-exome sequencing using a trio-based strategy [27], we designed functional experiments for genes within this cohort with rare (minor allele frequency MAF < 0.005) biallelic variants identified in two or more individual patients. Applying these criteria on this dataset, we identified three candidate genes— *ECE2*, *GPR98* and *PRPN13* (Table EV1). As a further filter, we limited our analysis to genes whose biallelic variants were assigned as potentially deleterious by any one of four *in silico* prediction

programs (Materials and Methods). Under this framework, only one gene—*ECE2*—remained.

*ECE2*, encoding for endothelin-converting enzyme-2 [27], is a metallopeptidase that has previously been associated with non-classical production of neuropeptides [34,35] and removal of intracellular amyloid-β in Alzheimer's disease [36,37], and its knockout in mouse leads to deficits in spatial learning and memory acquisition [38]. The function of this gene in the developing cortex, however, has not been described. The two patients with PH had biallelic missense variants in this gene that were *in silico*-predicted to be deleterious. Patient 1 (see example MRI images in Fig 1A) has two variants in *ECE2*, each inherited from a separate parent (c.655C>G, p.Arg219Gly; and c.2254C>T, p.Arg752Trp; RefSeq NM_001037324.2). Patient 2 is homozygous for a missense variant in this locus (c.2239T>A, p.Ser747Thr; RefSeq NM_001037324.2; Table EV1).

### ECE2 is enriched in human neurons

In order to verify that *ECE2* is expressed in human iPSC-derived models (Fig EV1A and B; for characterisation of our COs, see also supplementary figure 18 in Ref. [39]), we started with characterisation of *ECE2* mRNA expression in these model systems and detected higher *ECE2* expression in neurons compared to NPCs using qRT–PCR (Fig EV1C). This was confirmed for developing human brain by the online database libd (http://stemcell.libd.org/scb/) [40], containing publicly available RNA sequencing data from different stages of neuronal differentiation from 13 iPSC lines to neurons (Fig EV1D), and for primates in the neonate marmoset gene atlas [41]. Additionally, *in situ* hybridisation (ISH) for *ECE2*, performed in 50-day-old COs, confirmed higher levels in the cortical plate-like zone (CP′) than in the ventricular-like zone (VZ′; Fig 1B). Immunohistochemical (IHC) analysis for ECE2 also demonstrated higher protein levels in the CP′ (colocalising with MAP2$^+$ neurons) than in phospho-Vimentin$^+$ (pVIM$^+$) RG located in the VZ′ of COs, and an enrichment in the apical compartment of aRG was also evident (Fig 1C).

---

**Figure 1. Neuronal mislocalisation and migration defects found in individuals with *ECE2* variants are recapitulated in iPSC-derived *in vitro* models.**

A  Coronal and axial brain MRI of the patient with compound heterozygote mutations in *ECE2* shows nodules of heterotopic neurons lining the lateral ventricles.

B  *ECE2* expression on RNA level. *In situ* hybridisation for *ECE2* mRNA in 50-day-old cerebral organoids (COs) shows higher signal in the cortical plate-like zone (CP′; scale bar = 100 μm).

C  ECE2 expression on protein level. Immunohistochemistry (IHC) for ECE2 in 50-day-old COs shows accumulation in the CP′ and at the apical surface (scale bar = 50 μm).

D  Scheme showing the electroporation of DNA into ventricles of COs and the organisation of different cell types within the germinal zone. DNA is injected into the ventricle-like lumen and taken up by aRG′ via their apical processes. At 7 days post electroporation (dpe), the transfected construct can additionally be found in IP's and neurons upon differentiation of transfected aRG′ (green) (aRG′, apical radial glia; bRG′, basal radial glia; CP′, cortical plate; IP′, intermediate progenitor; IZ′, intermediate zone; SVZ′, subventricular zone; VZ′, ventricular zone).

E  COs transfected with microRNAs targeting *ECE2* (KD) or scrambled negative control (CTRL) and GFP and analysed 7 days later reveal an increase in ectopic neurons upon *ECE2* KD (transfected cells are shown in green, NEUN$^+$ neuronal nuclei in magenta; scale bar = 25 μm).

F  Graph depicting the number of ectopically located NEUN$^+$ cells in CTRL and KD-electroporated COs 7 dpe, in each ventricle normalised to the electroporated radial units, using two different microRNAs targeting *ECE2*. Data shown as box plot (mean = red line, median = black line, box represents 25$^{th}$ and 75$^{th}$ percentiles, whiskers extend to 10$^{th}$ and 90$^{th}$ percentiles, all outliers are shown; $n$ = number of ventricles analysed; ***$P$ < 0.001 in the Kruskal–Wallis one-way ANOVA on ranks and Dunn's pairwise multiple comparison).

G  Scheme depicting the experimental set-up of neuronal differentiation and time-lapse imaging under treatment with the ECE2-inhibitor PHOS [42].

H  Graphs showing different parameters of neuronal migration dynamics, tracked via live imaging. Upon ECE2 inhibition, significant decrease in velocity, increase in resting TP and increase in tortuosity are observed. Data shown as *z*-scores (CTRL, DMSO control; PHOS, phosphoramidon; Resting TP, resting time points; $N$, number of individual neurons analysed from two batches; Resting TP: *$P$ = 0.024; tortuosity: *$P$ = 0.035, ***$P$ < 0.001 in two-tailed chi-square test).

Data information: (B, C, E) Ventricle-like lumen in COs is marked as V′. See also Fig EV1.

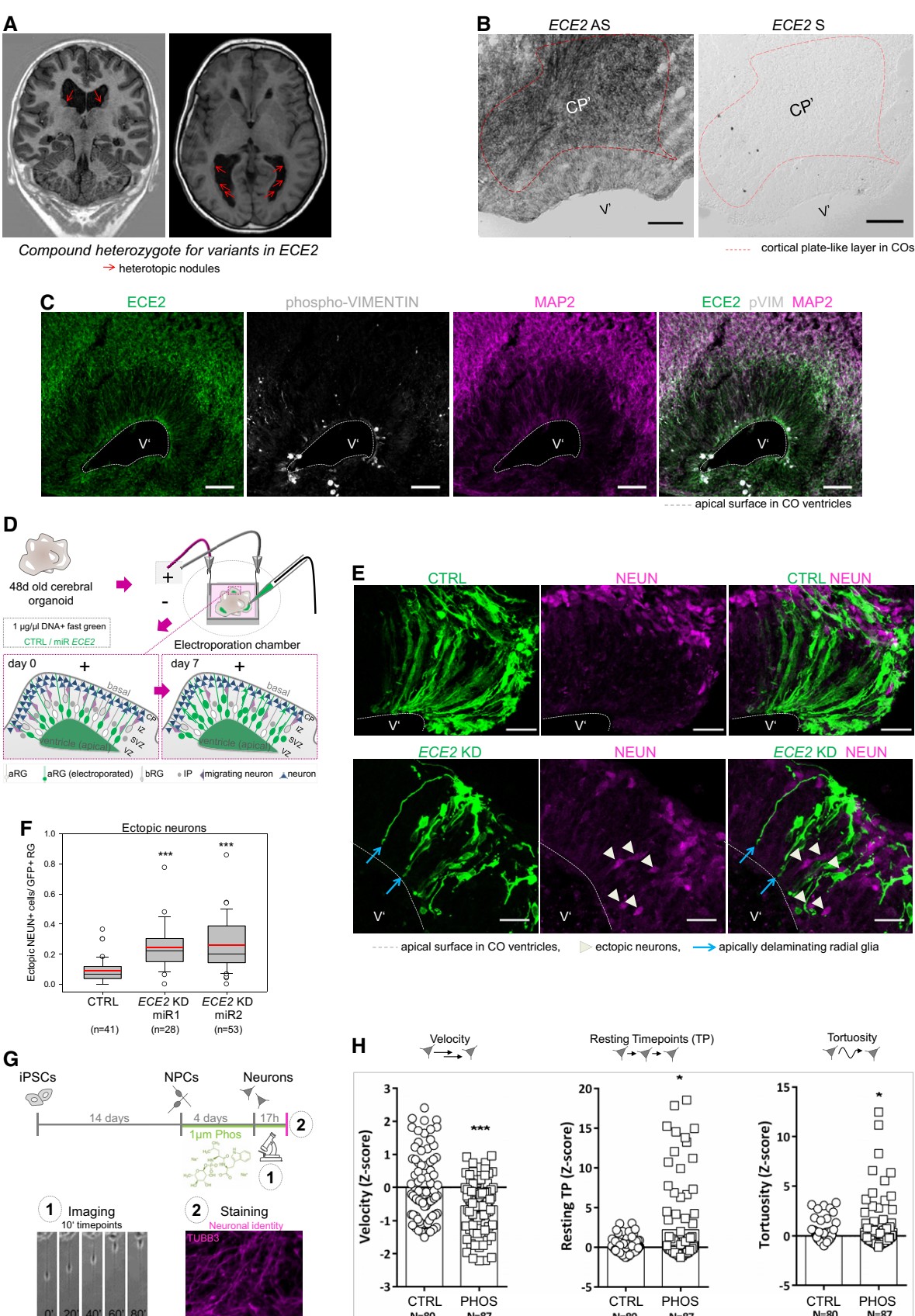

**Figure 1.**

## Knockdown of *ECE2* leads to accumulation of neurons in the VZ of COs

ECE2 was enriched within neurons; thus, to unravel the potential function of this protein in these cells, we performed acute knockdown (KD) of *ECE2* in human COs. To this end, plasmids expressing microRNAs (miRNAs) targeting the peptidase (Fig EV1E) and co-expressing GFP were generated using the BLOCK-iT system from Invitrogen and were injected into the ventricle-like lumen of 48-day-old COs prior to electroporation (Fig 1D). GFP$^+$-transfected aRGs and their progeny were analysed 7 days postelectroporation (7 dpe). At this stage, due to the organisation of COs into separated zones containing progenitors and neurons, changes in morphology, cell fate and migration can be analysed at once. Interestingly, a significant increase in differentiated NEUN$^+$ neurons, which were ectopically located within the VZ' after *ECE2* KD compared to control, was observed, mimicking some of the neuronal migration defects, e.g. ectopic neurons, observed in patients with PH (Fig 1E and F). Interestingly, most ectopically located neurons were not electroporated (NEUN$^+$GFP$^-$), but they were localised in close proximity to the transfected aRGs (arrowheads in Fig 1E). Electroporated aRGs had often lost their attachment to the apical surface (arrows in Fig 1E). This morphological change in transfected aRGs is already visible 4 dpe upon *ECE2* KD in COs (Fig EV1F, arrows).

## Inhibition of ECE2 activity changes the dynamics of neuronal migration *in vitro*

To assess whether the enzymatic function of ECE2 is also directly involved in neuronal migration, time-lapse imaging of young migrating neurons was performed upon inhibition of ECE2 activity. To this end, NPCs were cultured under differentiating conditions in the presence or absence of phosphoramidon [42] (PHOS; scheme in Fig 1G). This compound was shown to inhibit the function of ECE2 at low concentrations of 1 μM [42]. Chronic treatment (14 days) with PHOS did not increase cell death as revealed by cleaved caspase-3 staining (Fig EV1G). Young neurons cultured in the presence or absence of PHOS were imaged over 17 h, and several parameters, such as speed of migration, resting time points (TPs) and tortuosity, were analysed (Fig 1H). After inhibition of ECE2 activity, neurons migrated significantly more slowly, rested more often and migrated in a more tortuous path compared to control neurons.

## Acute KD of *Ece2* in the developing mouse cortex leads to ectopic rosettes and nodules

In order to reveal the role of *Ece2* in the developing cerebral cortex *in vivo*, acute KD of this gene's expression was performed by *in utero* electroporation (IUE) of microRNA (and GFP) expression vectors targeting *Ece2* (Fig EV2A) in the ventricles of the developing telencephalon of mice at embryonic day 13 (E13). Brains were then analysed 1 dpe (E14) and 3 dpe (E16).

The distribution of the GFP$^+$-electroporated cells was analysed 1 dpe by dividing the developing cortex in five equal bins, bin1 corresponding to apical side, while bin5 corresponding to pial side of the cortex (Fig EV2B). Upon *Ece2* KD, significantly more transfected (GFP$^+$) cells localised to bin2 (Fig 2A). Ectopic, basally located Pax6$^+$ progenitors were identified, which, compared to control, tended to localise more to bin2 than bin1 (Figs 2B and C, and EV2C), and were also positive for the active cell cycle markers Ki67 and phospho-Vimentin (Fig EV2D and E), with reduced apical Arl13b (Fig EV2F). This redistribution of Pax6$^+$ progenitors was not met with a significant change in neuron localisation 1 dpe, although some ectopic neurons were detectable at this early stage with a trend to increased localisation to bin3, and not bin4 and bin5 as in the control (Fig EV2G–I). Interestingly—likewise the results found in COs—ectopic progenitors were often not transfected (GFP$^-$; Fig 2C), Similarly, the morphology of transfected aRG was changed to a less radial and bipolar shape (Figs 2B and EV2D–F).

Analysis at 3 dpe (E13–E16; Figs 2D–J and EV2J–J′) revealed ectopic clusters, containing delaminated non-transfected Pax6$^+$ (Fig 2D) and Ki67$^+$ progenitors (Fig EV2J), accumulating as "rosettes" in the parenchyma around an Arl13b$^+$ area, surrounded by thickened processes of transfected cells (Fig 2E and F, blue and pink arrows). In contrast, we often found a reduction in apical Arl13b and β-catenin (Ctnnb1) in the electroporation site (Fig 2E and G, white arrowheads). Pax6- areas were also found in the progenitor zone underneath the ectopic rosette both at 1 dpe and at 3 dpe (circles in Fig 2B and D).

At 3 dpe, ectopic rosettes were surrounded by clusters of (mostly non-transfected) ectopic neurons, with both deep and superficial layer identities, often forming "nodules" (Figs 2H and I, and EV2J–J′). Moreover, single neurons were also more frequently localised to the apical surface of the lateral ventricles after *Ece2* knockdown compared to control cortices (Fig 2H). At both analysed TPs, ectopic rosettes and nodules contained almost

**Figure 2. Acute *Ece2* KD in the developing mouse cortex causes ectopic rosettes and nodules.**

A    Quantification of distribution of GFP$^+$-transfected cells at 1 dpe upon CTRL and *Ece2* KD by IUE (*n* = 3 CTRL and 4 *Ece2* KD brains; *P = 0.044 in one-way ANOVA and Tukey's multiple pairwise comparison; data shown as mean ± SEM).

B, C    *Ece2* KD leads to ectopic non-transfected Pax6$^+$ progenitors at 1 dpe (E13–E14). (B) IHC staining of GFP (transfected cells) and Pax6 at 1 dpe showing ectopic Pax6$^+$ progenitors (red circle) and a reduction in their apical localisation (yellow circle). (C) High magnification images of GFP and Pax6 showing that ectopic progenitors are mostly not transfected.

D–J    E13–16 *Ece2* KD. (D) GFP-Pax6$^+$ progenitors delaminate upon *Ece2* KD (red arrows), leaving behind a region free of Pax6$^+$ cells in the VZ (yellow circle) and forming ectopic rosettes (yellow box). (E, G) Reduction in apical Arl13b and β-catenin in the electroporation site (white arrowheads) indicates loss of apical belt integrity and disruption of apico-basal polarity upon *Ece2* KD. (F) Inside the nodule, which is surrounded by thickened GFP$^+$ processes (pink arrows), Arl13b staining shows the formation of an ectopic apical surface (blue arrow; enlargement of boxed area from D and E). (H) Deep layer and (I) upper layer neurons localise next to ectopic rosettes, forming nodules of ectopic neurons (blue arrowheads for single ectopic neurons and blue circle for large nodule). (J) Quantification at 3 dpe, counting the proportion of brains with reduced density of Pax6 in the VZ, ectopic progenitors or neurons, and changes in apico-basal polarity (including ectopic apical surface formation and reduction in apically localised apical markers; *n* = 6 brains per condition; exact binomial test: ****P < 0.0001).

Data information: Scale bars in (A, B, D, E, H and I) = 100 μm; scale bar in (C, F and G) = 50 μm. V marks the position of the ventricle. See also Fig EV2.

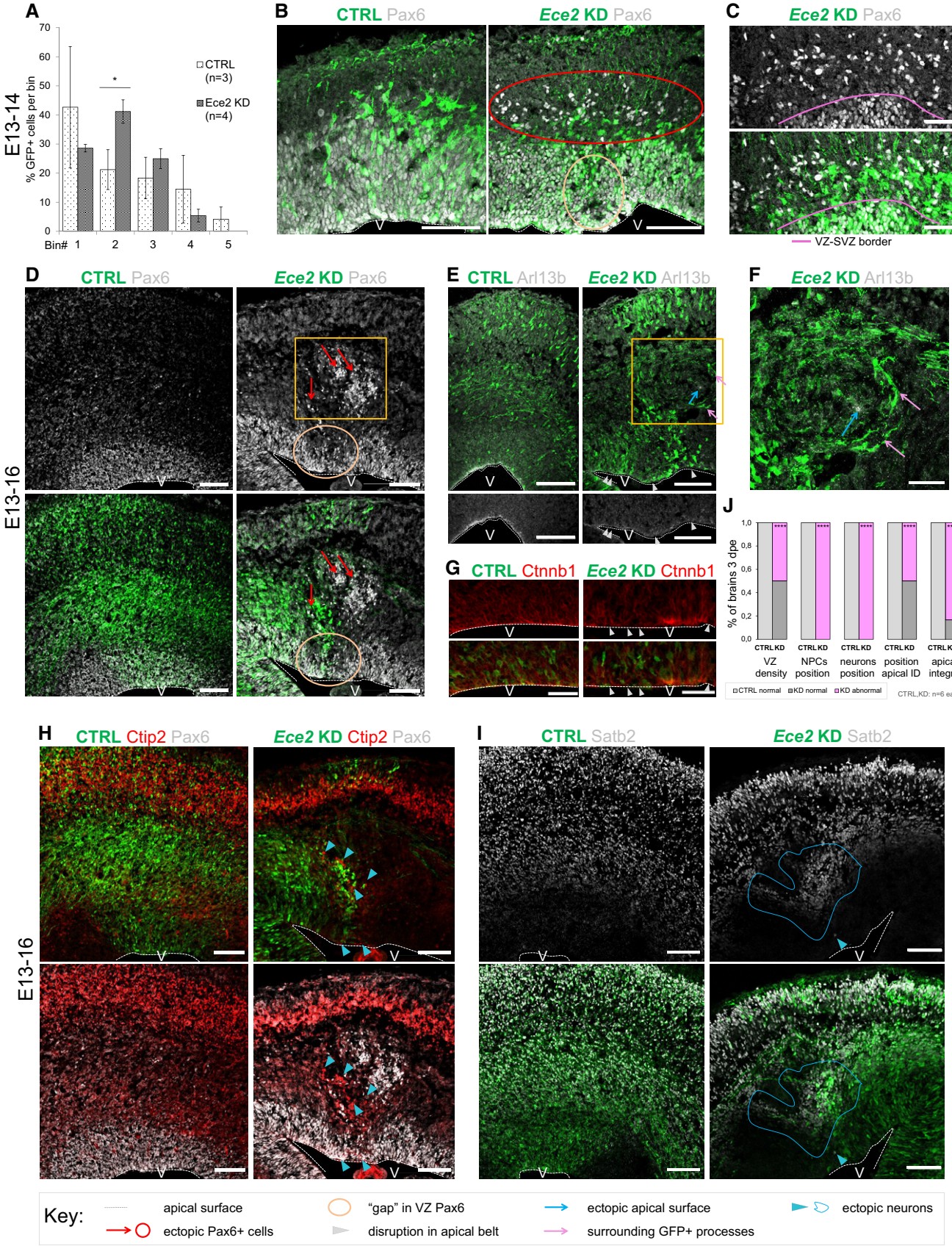

**Figure 2.**

exclusively non-transfected cells, but were surrounded by GFP$^+$ processes of transfected cells (Fig 2E and F). We consistently observed defects in the position and morphology of progenitors and neurons in all brains, but the severity of the phenotype strongly depended on the size and position of the electroporated area. Upon acute KD of *Ece2*, we identified significantly more ventricles with reduced Pax6 in the electroporation site, more ectopic rosettes and nodules, more ectopic Arl13b$^+$/β-catenin apical surfaces and a reduction in apical Arl13b and β-catenin (Fig 2J).

Taken together, the RNAi-mediated downregulation of *ECE2/Ece2 in vitro* and *in vivo* leads to mislocalisation of neurons in the developing cortex.

## Manipulation of ECE2 levels and activity reveal a role of ECE2 in cell-fate determination

To address the functional role of ECE2 in human cells, two parallel approaches were taken: COs were treated with the ECE2 inhibitor PHOS prior to IHC or fluorescence-activated cell sorting (FACS) analysis (scheme in Fig EV3A; Fig EV3B and C), and *ECE2* KO iPSCs and COs were generated and analysed (Figs 3 and EV3D–I).

Chronic treatment (21 days) of 60-day-old COs with PHOS leads to a change in their cellular composition: FACS analysis of NPCs (SSEA1- and BLBP-sorted cells) and of neurons (MAP2-sorted cells) revealed an increase in NPCs and a reduction in neurons (Fig EV3B and C).

CRISPR/Cas9-mediated genome editing in iPSCs targeting the *ECE2* locus resulted in a frameshift and premature STOP codon in one allele and fusion of first half of exon 3 and second half of exon 4 due to the loss of 80 AA in the second allele. The absence of *ECE2* mRNA in *ECE2* KO iPSCs was confirmed by qPCR (Fig EV3D). COs were generated from *ECE2* KO and the isogenic control iPSCs (Fig 3A) [32], and the absence of *ECE2* mRNA from *ECE2* KO COs was confirmed by ISH (Fig 3B). An initial delay in differentiation was visible during CO generation, with a later emergence of neuroepithelial tissue in the *ECE2* KO COs (Fig EV3E), without a significant difference in CO size compared to control (Fig EV3F). *ECE2* KO COs generated normal cortical-like regions (Figs 3A and EV3G).

Sixty-day-old COs were analysed by FACS using the progenitor marker PAX6, the mitotic marker PH3 and the neuronal marker DCX (see Fig EV3H for the gating strategy). Similar to the above-described findings for ECE2 inhibition, the proportion of progenitors was increased and that of neurons was decreased in *ECE2* KO COs compared to control (Fig 3C). This change in cellular proportions

was also confirmed by IHC, where we found that the number of PH3$^+$ cycling progenitor cells (normalised by the length of apical surface) was significantly increased in germinal zones of *ECE2* KO COs (Fig 3D and E), whereas the thickness of the MAP2$^+$ neuronal layer was decreased (Fig EV3I).

Altogether, both chronic enzymatic inhibition and the permanent KO of *ECE2* in COs lead to changes in the proportions of progenitor and differentiated cells.

Fluorescence-activated cell sorting analysis performed at later stages (107 days) confirmed the consistent significantly lower number of DCX$^+$ neurons in *ECE2* KO COs (Fig EV3J). On the contrary, the number of progenitors, quantified by PAX6 and PH3, was comparable to control COs (Fig EV3J).

Further exploiting the spatial organisation and separation of progenitor and CP-like zones in COs, we identified ectopic neurons within these COs in the VZ' after *ECE2* KO (Fig 3F). Some MAP2$^+$ neuronal processes were present throughout the progenitor zone in KO COs, and even though these structures were also observed in control COs, there was a clear significant increase in brain-like regions with progenitor zones containing neuronal cell bodies upon *ECE2* KO, with a concomitant decrease in normally organised ventricles in the KO condition (Fig 3G).

## ECE2 inhibition and KO lead to changes in actin and microtubule cytoskeleton

Due to the 3-dimensional organisation of germinal zones, COs represent a good system to analyse the cellular morphology, the integrity of cytoskeletal proteins and markers of apico-basal polarity after acute *ECE2* KD *in vitro* and *in vivo*.

Cytoskeletal dynamics and apico-basal polarity are essential both for the integrity of the RG scaffold used by locomoting excitatory neurons and for neuronal polarisation and migration itself [43]. These were thus characterised both in *ECE2* KO and in PHOS-treated COs to assess a possible involvement of ECE2.

A significant decrease in total actin was detected in *ECE2* KO COs (Figs 4A and EV4A), but interestingly, this change mostly affected the filamentous actin (F-actin) as shown by the increase in the ratio of globular (G)-actin to F-actin (Figs 4B and EV4B). Specifically, an increase in thickness of the F-actin-enriched apical adherens belt was detected both in COs treated with PHOS for 2 weeks (Fig EV4C and D) and in *ECE2* KO COs (Fig 4C and D). In addition, the apical F-actin-enriched honeycomb structure was also analysed. Here, we identified morphological alterations in neuroepithelial regions of *ECE2* KO COs (Fig 4E) consistent

---

**Figure 3.** *ECE2* KO and enzymatic inhibition cause a change in cell-fate and neuronal positioning in COs.

A   Brightfield images of 60-day-old CTRL and *ECE2* KO COs show normal formation of neuroepithelial structures.

B   ISH for *ECE2* confirms the absence of *ECE2* mRNA in 60-day-old *ECE2* KO COs (pink dotted line indicates position of the CP').

C   FACS analysis of 60-day-old *ECE2* KO COs reveals an increase in progenitors and decrease in neurons ($n$ = 7–8 samples of three pooled organoids each from three different batches; **$P < 0.01$; ***$P < 0.001$ in one-way ANOVA and Tukey's pairwise multiple comparison).

D   Example images of IHC for PH3$^+$ cells in 60-day-old CTRL and *ECE2* KO COs.

E   Quantification of PH3$^+$ cycling cells in 60-day-old COs, normalised to length of the apical surface of ventricle-like structures, shows an increase upon *ECE2* KO (data shown as box plots with median as black line, mean as red line; $n$ = number of analysed ventricles from two batches of COs; one-way ANOVA: **$P = 0.004$).

F, G   In *ECE2* KO COs, more ventricles are disorganised in terms of neuronal localisation as revealed from MAP2$^+$ cell bodies in the progenitor zone ($n$ = number of ventricles from two independent batches; two-tailed Mann–Whitney $U$-test ($U$ = 458) and chi-square test: $\chi^2_{(2)}$ = 25.42, ****$P < 0.0001$).

Data information: (B, D, F) Scale bars = 50 μm; ventricle-like structures in COs are marked with V'. See also Fig EV3.

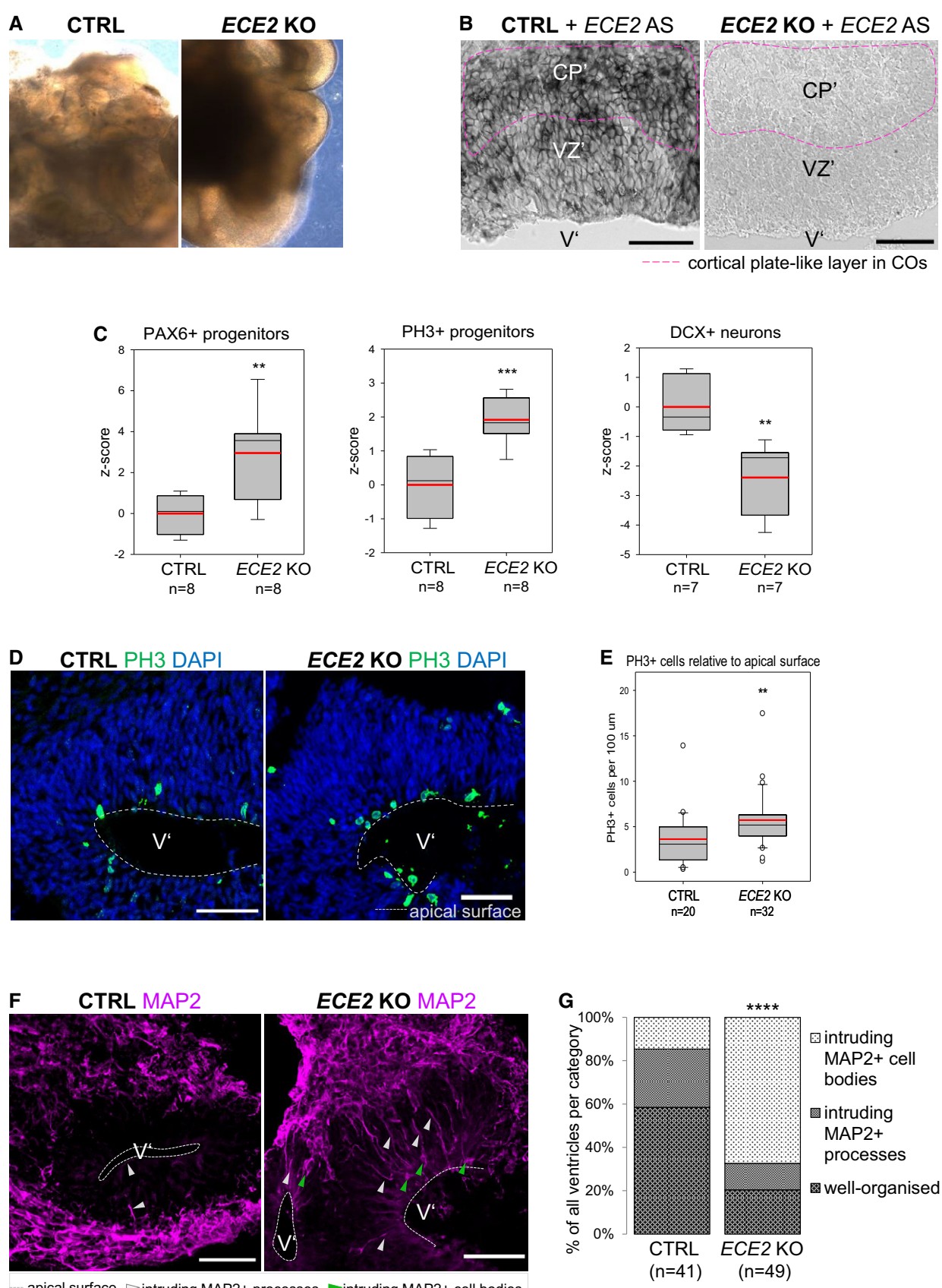

**Figure 3.**

with a loosening of the apical junctions underlying the increased thickness.

Next, we analysed the stability of microtubules as the second important component of the cytoskeleton. We used IHC for acetylated tubulin (ac-tub), which is enriched in stabilised microtubules [44]. IHC of ac-tub in PHOS-treated and *ECE2* KO COs revealed some normal germinal zones and some with a reduction and/or disorganisation in stabilised microtubules. In particular, signal intensity of ac-tub was measured in neuroepithelial structures and showed a significant reduction upon chronic PHOS treatment (Fig EV4E and F) and in *ECE2* KO COs (Fig 4F and G).

We next assessed apico-basal polarity in PHOS-treated and *ECE2* KO COs by IHC for ARL13B as a marker for the microtubule-based

primary cilium that contacts the ventricular lumen in aRG. We found again some normal ventricles and some with reduced or totally absent localisation of ARL13B at the apical surface (Figs EV4G and 4E and H). The number of ventricles with normal ARL13B was significantly reduced (Figs EV4H and 4I).

To further support our finding of reduction in apically localised ARL13B and to minimise artefacts due to cutting orientation of COs, we used iDISCO [45] of PHOS-treated whole COs.

Whole-mount IHC for ARL13B, tissue clearing and imaging of whole COs confirmed the reduction in apical ARL13B upon ECE2 enzymatic inhibition (Movies EV1 and EV2).

Finally, to visualise aRG morphology upon *ECE2* KO, we electroporated GAP43-GFP plasmid into ventricle-like structures of CTRL

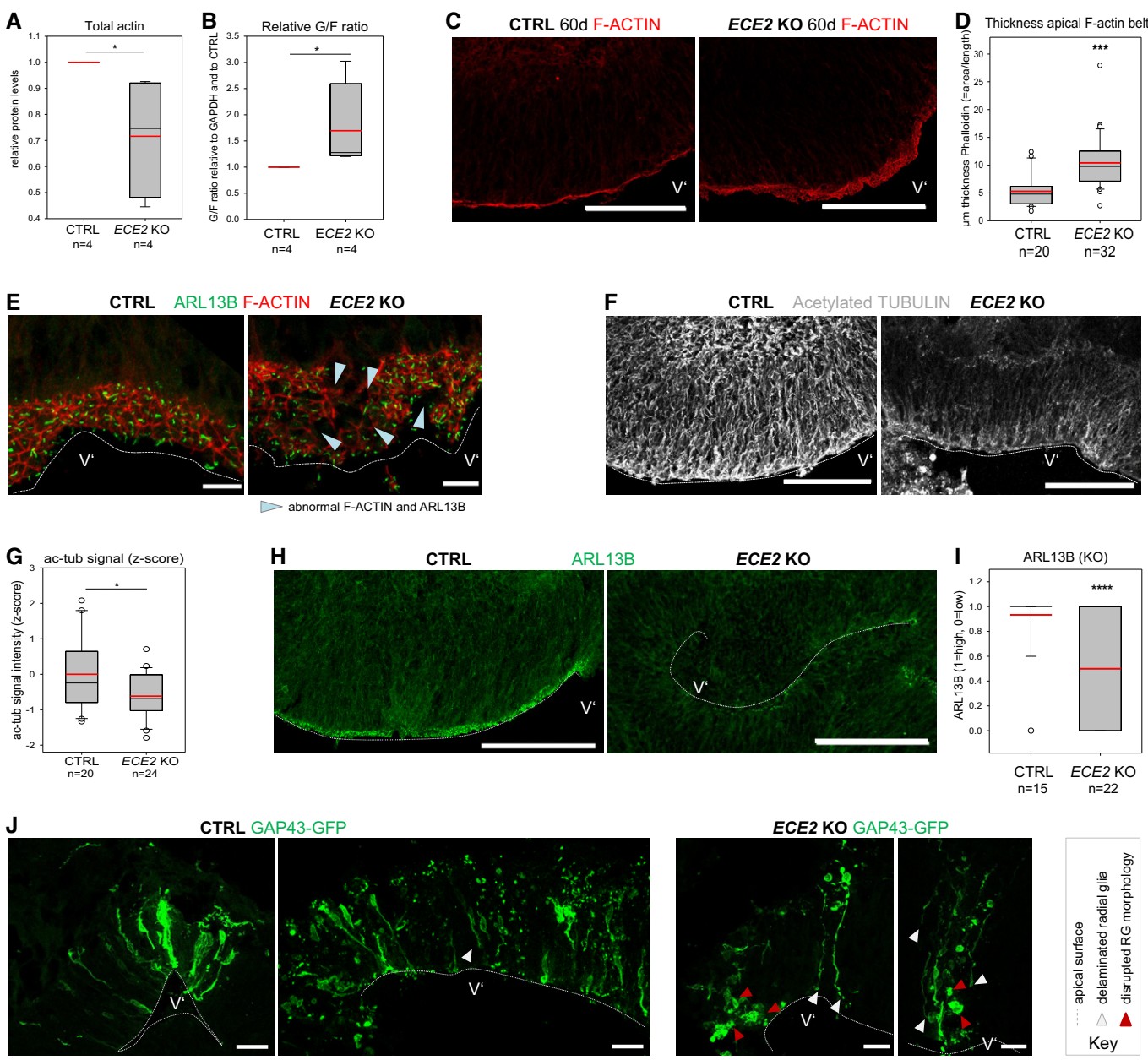

Figure 4.

**Figure 4.** *ECE2* KO affects neural progenitor polarity and actin and microtubule cytoskeleton in COs.

A    Quantification of actin levels from Western blot of whole CO lysate reveals a reduction in *ECE2* KO COs (n = 4 independent batches of CTRL and *ECE2* KO COs with three pooled COs each sample; *P = 0.021 in the Kruskal–Wallis one-way analysis of variance on ranks and Dunn's pairwise multiple comparison).

B    Fractionation of G-actin and F-actin from CTRL and *ECE2* KO COs and analysis via Western blot additionally reveal an increase in G- at the expense F-actin upon *ECE2* KO, suggesting a reduction in F-actin belt integrity (n = 4 independent batches of CTRL and *ECE2* KO COs with three pooled COs each; *P = 0.021 in Kruskal–Wallis one-way analysis of variance on ranks and Dunn's pairwise multiple comparison).

C, D   The thickness of the apical F-actin belt is increased in *ECE2* KO COs. (C) Example images of apical F-actin belt in 60-day-old CTRL and *ECE2* KO COs with F-actin labelled by Alexa Fluor 594-conjugated Phalloidin (scale bar = 100 μm). (D) Quantification of the thickness of apical F-actin belt in Fiji [82] by measuring the area of F-actin and dividing by the length of apical surface reveals significant increase in *ECE2* KO COs (box plots: mean = red line, median = black line, box represents 25th and 75th percentiles, whiskers extend to 10th and 90th percentiles, all outliers are shown; n = number of analysed ventricles in two batches; Kruskal–Wallis one-way ANOVA on ranks and Dunn's pariwise multiple comparison: ***P < 0.001).

E    IHC for ARL13B and F-actin in 60-day-old CTRL and *ECE2* KO COs shows disruption of honeycomb-like structure of the apical adherens belt and of apico-basal polarity in terms of apically localised primary cilium in *ECE2* KO COs (scale bar = 10 μm).

F, G   The microtubule cytoskeleton is changed in COs upon *ECE2* KO. (F) Example images of ac-tub IHC in CTRL vs. *ECE2* KO COs (scale bar = 100 μm). (G) Quantification of ac-tub in ventricles of *ECE2* KO COs by measurement of the mean grey value in Fiji shows significant reduction upon *ECE2* KO (n = number of analysed ventricles; *P = 0.019 in one-way ANOVA and Tukey's pairwise multiple comparison).

H, I   Apico-basal polarity is impaired upon ECE2 KO as visible in example images of ARL13B IHC in CTRL vs. KO COs (H; scale bar = 100 μm). (I) Quantification of germinal zones for normally high ("1") vs. reduced ("0") apical ARL13B reveals a reduction in the proportion of normal ventricles in the absence of ECE2 (n = number of analysed ventricles from two batches; ****P < 0.0001 in exact binomial test).

J    Analysis of CTRL and *ECE2* KO COs at 1 dpe with GAP43-GFP reveals RG that delaminated (white arrowheads) and/or lost their bipolar morphology (red arrowheads) in the KO COs (scale bar = 25 μm).

Data information: Ventricle-like lumen is marked by V'. See also Fig EV4.

and KO COs to label the membrane of transfected aRG. Analysis at 1 dpe revealed that in *ECE2* KO COs, aRG can lose their apical attachment (which was still present during electroporation for take-up of the construct) or even completely lose their bipolar morphology (Fig 4J).

Altogether, we identified changes both in the microtubule and actin cytoskeleton and in the polarity of aRG upon chronic inhibition or permanent genetic deletion of *ECE2*.

Despite the numerous processes that ECE2 is involved in, the molecular role of *ECE2*—causing both cell-autonomous and non-cell-autonomous defects upon its disruption during neurogenesis—still remains undefined.

### Forced expression of *ECE2* results in neuronal mispositioning

In order to understand whether the phenotype observed in patients is due to gain or loss of function of *ECE2,* we manipulated the levels of *ECE2* by forcing its expression in the developing mouse cortex (Fig 5A–F) and in human COs (Fig 5G–L). Surprisingly, overexpression (OX) of *ECE2* in the mouse cortex resulted in a very similar effect on NPC and neuron localisation as did *Ece2* KD, namely redistribution of GFP$^+$Pax6$^+$ progenitors and total Ctip2$^+$ neurons (already 1 dpe; Fig 5A–C). This loss of correct positioning of progenitors (Sox2$^+$) and migration of newly generated Ctip2$^+$ and Satb2$^+$ neurons was clearly more evident 3 dpe (Fig 5D–F with blue arrowheads and circles and Fig EV5A and B).

Upon OX of *ECE2* in COs, the direct effect of *ECE2* on the progenitors' proliferation and neuronal differentiation observed in *ECE2* KO COs was confirmed as acute OX of *ECE2* for 7 days led the opposite phenotype observed in *ECE2* KO and PHOS conditions. Namely, a decrease in total PH3$^+$ progenitors per electroporated surface length (white arrowheads in Fig 5G) and an increased number of GFP$^+$NEUN$^+$ neurons were observed (yellow arrowheads in Fig 5G) (Fig 5G–I). Similar to *in vivo,* in COs a cell-non-autonomous mislocalisation of neurons in the VZ' arose upon OX of *ECE2* (Fig 5G and J) and resembled that observed upon acute *ECE2* KD.

Importantly, acute OX of *ECE2* in *ECE2* KO COs restored local physiological levels of *ECE2* and resulted in a rescue of the number of ectopic neurons residing in the electroporated progenitor zone (Fig 5K and L, blue arrowheads) and of the morphology of electroporated aRG (Fig 5K).

### Proteomics reveals ECE2 involvement in extracellular matrix production and integrity

To gain a deeper insight into the mechanisms underlying the observed phenotypes in *in vivo* and *in vitro* models, we performed whole-proteome analysis of 60-day-old CTRL and *ECE2* KO COs. Mass spectrometry-based proteomics has recently been used to compare trisomy 21 patient and control-derived COs and revealed (i) that the neurodevelopmental trajectory shown several times by scRNAseq [46–48] is also recapitulated on the protein level in COs, (ii) a number of dysregulated proteins in trisomy 21 and (iii) the reversal of dysregulated proteins by drug treatment [preprint: 49]. Here, quantitative proteome analysis of whole-organoid lysate revealed a number of significantly up- and downregulated proteins upon *ECE2* KO (Fig 6A and Dataset EV1). These included upregulation of PAX6 and downregulation of MAP2, confirming FACS data from *ECE2* KO and PHOS-treated COs (Figs 3 and EV3). Additionally, the identified downregulated proteins included numerous proteins with functions that are highly relevant to cortical development and its disorders. Figure 6B illustrates a classification of all downregulated proteins into the following classes (see colour code in Table EV2) [50–52]. These included genes/proteins whose disruption is known to cause PH (e.g. FLNA [14–17]) or other disorders of neurodevelopment. A number of additional microtubule and microtubule-associated proteins (MAPs) were also downregulated in *ECE2* KO COs, strengthening our previous findings (Figs 4 and EV4), as were additional actin-associated/actin-organising proteins, such as ezrin and moesin, which are interactors of RHOA and FLNA (Fig EV5C).

In addition, guidance and adhesion factors such as SLITs and FLRTs were enriched in the set of downregulated proteins in *ECE2* KO COs.

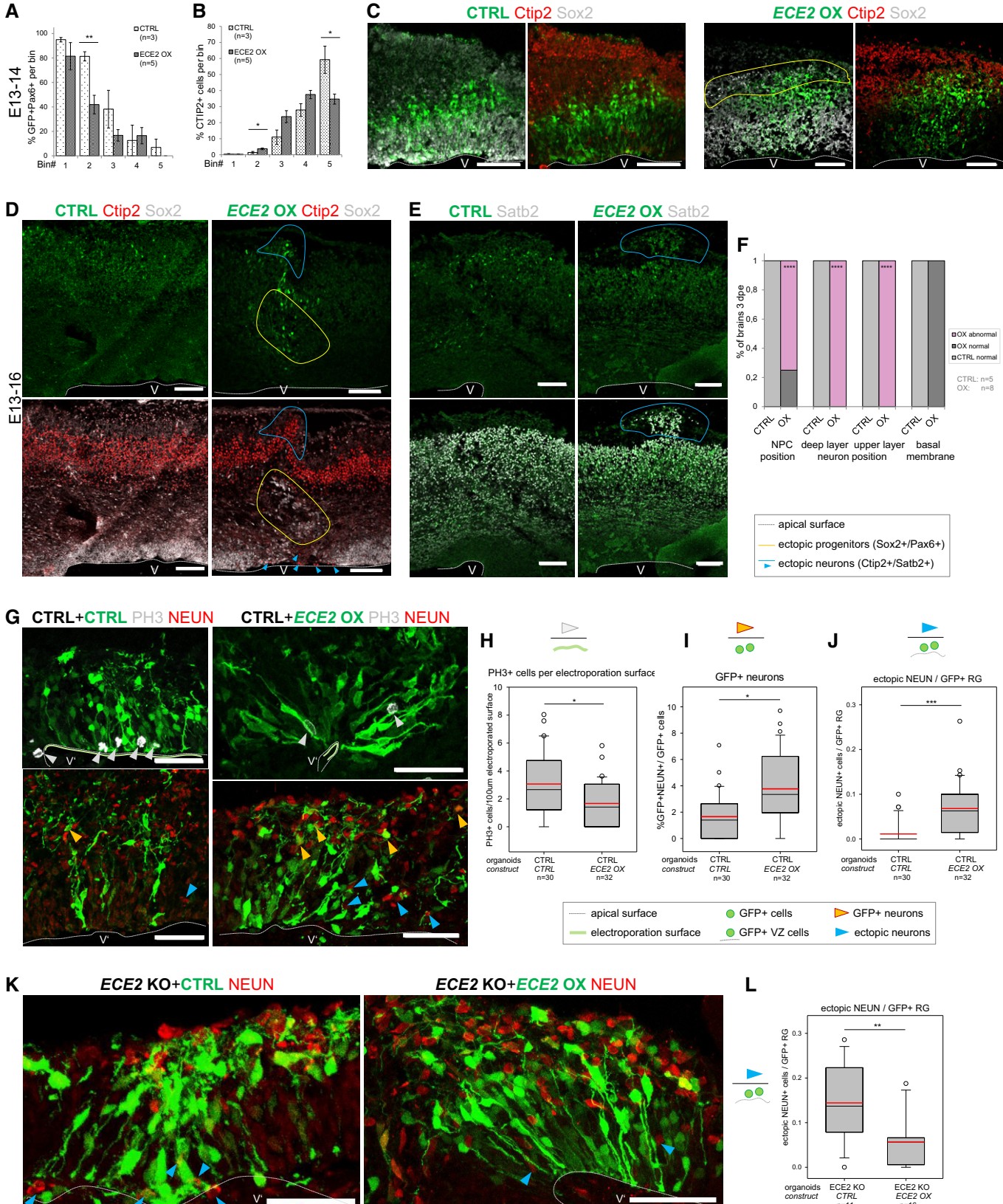

**Figure 5.**

**Figure 5.  Forced expression of *ECE2* in the developing mouse brain and in COs causes mislocalisation of neural progenitor cells and neurons and rescues neuronal positioning in *ECE2* KO COs.**

A, B   Binning analysis at 1 dpe after forced expression of *ECE2* in the developing mouse cortex reveals ectopic positioning of Pax6+GFP+ neural progenitors with significantly reduced localisation to bin2 (A) and increased localisation of Ctip2+ deep layer neurons to bin2 with reduction in bin5 (B) (for binning strategy, see Fig EV2B; *n* = 3 CTRL and 5 *ECE2* OX brains E13–14; data shown as bars with mean ± SEM; **P* < 0.05 and ***P* < 0.01 in Student's *t*-test).

C   Example images E13–14 of IHC for Pax6 and Ctip2 upon CTRL and *ECE2* OX conditions (scale bar = 100 μm; electroporated cells shown in green).

D–F   E13–16 images and analysis of *ECE2* OX in the developing mouse cortex reveal ectopic positioning of Sox2+ (D) progenitors and Ctip2+ deep- (D) and Satb2+ upper layer neurons (E) (scale bars = 100 μm; electroporated cells shown in green). (F) Quantification at 3 dpe, counting the proportion of brains with normal *vs.* abnormal positioning of Sox2+ neural progenitors and Ctip2+ and Satb2+ neurons and that of brains with intact vs. disrupted basal membrane (*n* = 5 CTRL and 8 ECE2 OX brains; exact binomial test: *****P* < 0.0001).

G–L   *ECE2* OX in COs (electroporated cells shown in green; scale bars = 50 μm; *n* = number of CO ventricles analysed from two batches). (G) Example images of COs 7 days after CTRL or *ECE2* OX electroporation and IHC for PH3+ mitotic cells and NEUN+ neuronal nuclei. (H) Quantification of PH3+ cells relative to electroporated apical surface length reveals a reduction upon *ECE2* OX in CTRL CO background (**P* = 0.046 in Kruskal–Wallis one-way analysis of variance on ranks and Dunn's pairwise multiple comparison). (I) Quantification of GFP+ neurons as proportion of all GFP+ cells reveals an increase upon *ECE2* OX in COs (**P* = 0.021 in Kruskal–Wallis one-way analysis of variance on ranks and Dunn's pairwise multiple comparison). (J) Quantification of ectopic neurons localising to the VZ' relative to electroporated radial units (GFP+ cells in the VZ') reveals a significant increase upon *ECE2* OX (****P* < 0.001 in Kruskal–Wallis one-way analysis of variance on ranks and Dunn's pairwise multiple comparison). (K, L) Forced expression of *ECE2* in *ECE2* KO COs rescues neuronal positioning. (K) Example images of CTRL- and *ECE2*-electroporated *ECE2* KO COs at 7 dpe. (L) Quantification of ectopic neurons relative to electroporated radial units reveals a decrease upon *ECE2* OX in *ECE2* KO COs, thus rescuing the neuronal positioning phenotype in the absence of *ECE2* OX (***P* = 0.002 in Kruskal–Wallis one-way analysis of variance on ranks and Dunn's pairwise comparison).

Data Information: (H–J, L) Outliers exceeding mean ± 2 standard deviations were removed. (C–E) Ventricle marked by V. (G, K) Ventricle-like lumen marked by V'. See also Fig EV5.

The largest group of downregulated proteins, however, consisted of ECM components and their receptors, e.g. laminins, lumican, decorin and six different collagens (see Fig 6B and Table EV2).

Analysis of protein–protein interactions (PPI) of downregulated proteins, using String.db [52], showed an arrangement of downregulated proteins into a tightly interconnected network (Fig EV5D), and GO term analysis [51,52] revealed an enrichment of proteins with functions that are highly relevant to cortical development, such as adhesion, polarity and cytoskeleton (Fig 6B, Dataset EV2).

As ECE2 is a protease localised in secretory vesicles [53], the cell culture supernatant of COs was also analysed by mass spectrometry (MS) for differences in the "secretome" between *ECE2* KO and control COs. Two proteins were identified as significantly downregulated upon *ECE2* KO (see Fig 6C, Dataset EV3): Lumican and TGFBI, both are collagen-binding proteins (Table EV3). Additionally, other collagens in the secretome of *ECE2* KO COs were somewhat decreased (depicted in green in Fig 6C).

Microtubule dynamics is regulated by different types of MAPs, such as EML4, MAP2 and MAP4, which were downregulated in the proteome of *ECE2* KO COs (Dataset EV1 and Table EV2). Additionally, posttranslational modifications (PTMs) of tubulins and MAPs influence microtubule stability, the interaction capability of MAPs,

and microtubule-based transport [54]. To this end, we further investigated differentially phosphorylated protein sites from the whole-proteome analysis between CTRL and *ECE2* KO COs (Dataset EV4). Strikingly, we detected differences in several proteins involved in apical adhesion, cytoskeletal and motor functions, with special enrichment of microtubule-related proteins among exclusive phosphorylation sites (Dataset EV5) and of actin cytoskeleton-related proteins among differentially phosphorylated sites (Dataset EV6; Fig 6D).

Taken together, proteomics of CTRL and *ECE2* KO COs confirmed FACS and IHC data on ECE2's role in apical belt integrity, apico-basal cell polarity, and actin and microtubule cytoskeleton integrity and additionally revealed a function for ECE2 in cytoskeletal protein phosphorylation and—as identified both in CO lysate and in secretome—in the production of ECM proteins and their receptors. All identified phenotypes upon *ECE2* manipulation are summarised in Fig 6E–G.

## Discussion

In this work, we used a combination of *in silico* prediction tools with human *in vitro* and mouse *in vivo* experiments to identify and

**Figure 6.  Proteomics reveal a role for ECE2 in ECM production and integrity.**

A   Volcano plot visualises the proteomic analysis from 60-day-old CO lysates showing proteins with significantly lower and higher expression upon *ECE2* KO (*n* = 3 batches of 60-day-old CTRL and *ECE2* KO COs; FDR 0.05; s0 1; dashed lines indicating the cut-off; names of example proteins are shown).

B   Pie chart with proteins downregulated in *ECE2* KO COs sorted by relevant categories (Number in the elements = number of proteins falling into each category, with numerous proteins falling into several ones) [50–52].

C   Volcano plot of proteomic analysis of cell culture supernatant from 55-day-old CTRL and *ECE2* KO COs shows two significantly downregulated proteins (red) in the secretome (*n* = 3 batches of CTRL and *ECE2* KO COs; FDR 0.05; s0 1; dashed lines indicating the cut-off). Non-significantly changed proteins of interest are highlighted in green below the cut-off curves.

D   GO cellular component analysis of proteins with dysregulated phosphorylation sites (either found exclusively in CTRL or KO COs or up- or downregulated in *ECE2* KO COs relative to CTRL) identified in whole CO proteome analysis.

E–G   Schemes summarising identified phenotypes in *in vivo* and *in vitro* model systems upon acute *ECE2* KD (E) or chronic inhibition or KO (F) or upon acute *ECE2* OX (G) and thus the molecular action of ECE2 (aRG, apical radial glia; bRG, basal radial glia; CP, cortical plate; ECM, extracellular matrix; IP, intermediate progenitor; IZ, intermediate zone; pe, postelectroporation; SVZ, subventricular zone; VZ, ventricular zone.).

Data information: See also Fig EV5 and Tables EV1–EV3 and EV4–EV9.

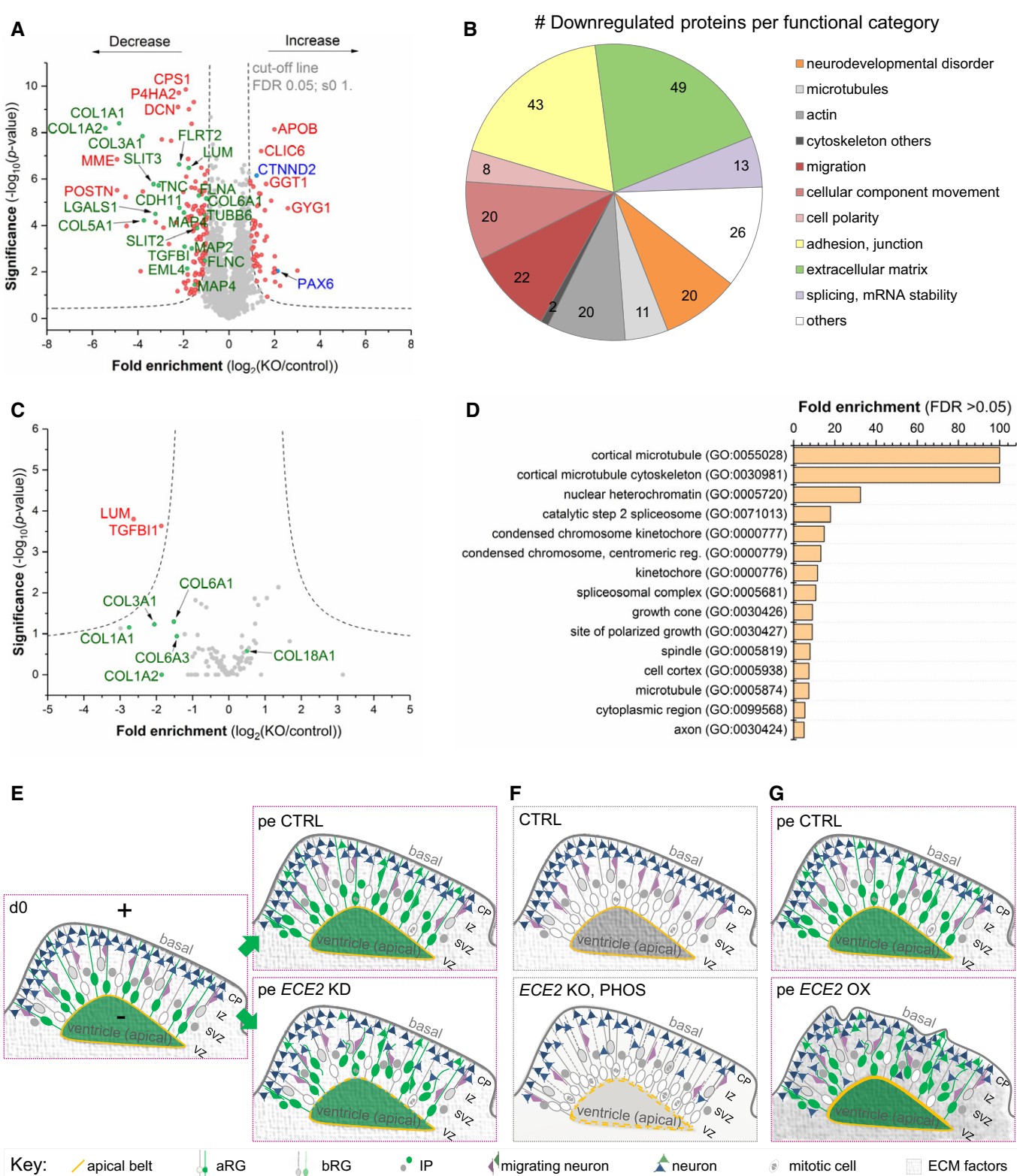

**Figure 6.**

characterise a novel candidate causative gene for the cortical developmental disorder PH. Based on the recently shown genetic heterogeneity of PH [27], we applied a stringent filtering strategy here, focussing on rare biallelic whole-exome sequencing variants (with one allele each inherited from a healthy parent) that were *in silico*-predicted to be deleterious. The function of the resulting candidate gene *ECE2* was in the following examined by manipulation of its expression and inhibition of the encoded enzyme's activity. Our

results suggest several mechanisms through which ECE2 plays a role in cortical development and therefore, together with other recent studies [28,29], underline the power of next-generation genetic analyses for the identification of responsible genes and the striking success of the applied *in silico* candidate selection theorem.

### *In vitro* and *in vivo* approaches reveal a role for ECE2 in neuronal positioning

Through the application of acute and permanent paradigms *in vitro* in combination with acute *in vivo* manipulation, we demonstrated here a role for *ECE2* in brain development; namely, acute *ECE2/Ece2* KD and OX lead to malpositioning of neurons. Moreover, in *ECE2* KO COs, generated by the permanent genetic deletion of *ECE2* in iPSCs, significantly more ventricles contained ectopically positioned neurons. This phenotype could be rescued by forced expression of *ECE2* in *ECE2* KO COs. The malpositioning of neurons characterising the brains of patients with PH is thus partially recapitulated both after acute *ECE2/Ece2* KD and in *ECE2* KO COs and resembles the phenotype detected in other CO models of PH [13,28,29].

### Cell-autonomous and non-cell-autonomous role of ECE2 in neuronal migration

Acute KD of *ECE2* in CO ventricles lead to a significant increase in ectopically located neurons, mimicking the phenotype of ectopic neurons, which localise to nodules of grey matter along the lateral ventricles in the brains of patients with PH. Interestingly, most of these ectopic neurons within the CO VZ's were not electroporated but were localised next to electroporated aRGs that had lost their apical attachment and showed a delamination phenotype. This suggests firstly a role of ECE2 in aRG and adherens junction integrity as suggested from the enrichment of the protein in the apical aRG compartment and secondly a non-cell-autonomous function of *ECE2* in neuronal migration, namely its role in the establishment/maintenance of the RG scaffold used by locomoting neurons [9–13]. This was further supported by the finding that, upon *ECE2* KD in COs, electroporated aRGs underwent morphological changes leading to a loss of their attachment to the apical surface, resulting in their delamination. Electroporation of GAP43-GFP plasmid in *ECE2* KO CO ventricles to visualise the aRG membrane further showed apical detachment and loss of apico-basal polarity.

In the developing mouse cortex, a similar phenotype was detected upon acute *Ece2* KD; namely, mostly non-electroporated Pax6$^+$ progenitors delaminated from the VZ and acquired ectopic positions, leaving behind a Pax6-sparse area in the VZ and forming basal rosettes around an ectopic apical surface. GFP$^-$ neurons of deep and upper layer identity mislocalised in close vicinity to them in nodule-like accumulations. These ectopic rosettes and nodules were encircled by thickened GFP$^+$ processes, indicating that also *in vivo*, non-cell-autonomous mechanisms may contribute to this redistribution of cortical progenitors and neurons. Just as in COs, electroporated aRG acquired a change in morphology upon *Ece2* KD *in vivo*, which was also visible in a loss of apically localised proteins. We therefore suggest that the PH-like disruption originated from a delamination of progenitor cells caused by a morphology and polarity change of aRG. These changes then lead to the formation of subapical clusters of progenitors that possibly generated

neurons ectopically. Although the final position of ectopic neurons differed between COs and mouse, these findings strongly suggest a non-cell-autonomous nature of the PH upon biallelic functional *ECE2* mutation.

In parallel to the non-cell-autonomous phenotype of ECE2 in neuronal migration, we have also observed an additional cell-autonomous function of ECE2 directly in neuronal migration by imaging neuronal migration behaviours *in vitro*. In particular, we cultured and time-lapse-imaged migrating neurons in 2D with PHOS at a concentration that had previously been shown to suppress ECE2 activity without noteworthy influence on ECE1 activity [42]. Time-lapse analysis revealed significant changes in different neuronal motility parameters. As the culture consisted of a mix of young neurons and NPCs that were sparsely plated without cell–cell contact, we can exclude any influence of the present NPCs or of neighbouring neurons on the neuronal behaviour. Therefore, we conclude that the activity of ECE2 is also cell-autonomously required for normal neuronal dynamics. Taken together, these data support a role of *ECE2* in the developing human brain.

### ECE2's role in cell-fate determination

Functional studies of ECE2's role in cortical development were conducted via two parallel approaches of chronic ECE2 inhibition in COs on the one hand and of *ECE2* KO CO generation on the other hand. Firstly, normal cortical-like regions were generated in *ECE2* KO COs, suggesting that *ECE2* deficiency is still permissive for the first steps of neurodevelopment. This is coherent with the phenotype of patients with PH, where only a subset of neurons fails to migrate properly and the cortex develops normally. The analysis of the cell composition after chronic enzymatic inhibition or the permanent KO of *ECE2* in COs (60 days old) revealed a change in the proportions of progenitors and differentiated cells with an increase in NPCs at the expense of neurons. These findings demonstrate a role for ECE2 in neurogenesis and suggest that the higher expression of *ECE2* in neurons may be required for neuronal differentiation. To confirm these findings, we electroporated CTRL COs with an *ECE2* OX construct and detected a reduction in mitotic cells and an increase in neurons produced. This underscores ECE2's function in neurogenesis. Further, MS-based whole-proteome analysis of CTRL and *ECE2* KO COs confirmed these data by detection of increased PAX6 and decreased MAP2 levels upon *ECE2* KO. Interestingly, analysis in older *ECE2* KO COs (107 days) revealed a reduction in neurons without a change in progenitors, suggesting that *ECE2* plays a role in proliferation only at early stage of development. A role for PH genes in the regulation of cell fate has also been demonstrated for other genes: patients with *ARFGEF2* mutations have microcephaly [21], and males with *FLNA* mutations that die at birth have thinner cortices [55]. Moreover, acute KD of *Fat4* and *Dchs1* in the developing mouse cortex, whose mutations cause PH in humans, leads to increased progenitor and reduced neuronal numbers and result in accumulation of ectopic neurons [10]. In COs derived from PH patients with *FAT4* and *DCHS1* mutations, the opposite phenotype was observed [13]. Also, *Cdc42* and *RhoA* disruptions cause changes in proliferation and brain size [56,57] and the recently discovered PH candidate gene *PLEKHG6* regulates NPC differentiation and neuronal migration via RHOA with its KD in COs phenotypically recapitulating PH [29]. Our findings highlight

the importance and value of a combination of acute manipulation methods (RNAi-mediated KD and forced OX) with chronic (long-term PHOS treatment) or permanent (genome editing) ones: due to its strong non-cell-autonomous action, the role of ECE2 in neurogenesis could not be identified via sparse acute transfection but only when affecting all cells in COs.

## Intracellular functions of ECE2: actin and microtubule cytoskeleton integrity

Based on the detected changes in neuronal positioning, aRG morphology and apico-basal polarity upon acute *ECE2* KD, actin and microtubule cytoskeleton were obvious candidates to analyse in our models. Both in KO and in chronically PHOS-treated COs, we detected a decrease in total actin and especially in F—compared to G-actin levels and a thickened apical F-actin belt with a disrupted, loosened honeycomb structure. ECE2 may thus directly or indirectly influence the regulation of the actin filament (re)organisation, contributing to the morphology and delamination of aRG in the developing cerebral cortex. The role of the actin cytoskeleton and its anchoring to apical adherens junctions has been described in several NMDs: conditional deletion of α-E-catenin disrupted the coupling of adherens junction proteins with intracellular actin fibres, leading to the loss of apico-basal polarity and an increased G/F-actin ratio, and eventually resulted in subcortical band heterotopia [58]. *Rapgef2* and *6* were also implicated in adherens junctions as (conditional) KO of one or both of them resulted in RG scaffold disruption and delamination, with ectopic RG and IPCs giving rise to a huge ectopic cortical mass [59]. Similarly, loss of the polarity regulator *Lgl1* in the mouse cortex disrupted adherens junctions, leading to ectopic neural progenitors with disorganised aRG fibres and ectopic neurons. Moreover, aRGs lacking Lgl1 showed hyperproliferation at the expense neuronal differentiation [60]. Conditional KO of *RhoA* destabilised both actin and microtubule cytoskeleton in RG with an increase in G/F-actin ratio. Although neurons lacking *RhoA* would migrate normally, these RG defects produced subcortical band heterotopia via non-cell-autonomous mechanisms [9]—similar to what we show here for acute *ECE2/Ece2* KD. We additionally analysed stabilised microtubules and found a reduction both in PHOS-treated and in *ECE2* KO COs. The number of ventricles with normal apico-basal polarity, analysed based on apical ARL13B signal, was also reduced in sectioned and whole-mount cleared COs. Interestingly, an involvement of primary cilium abnormalities in ectopic neuronal positioning was recently suggested from KD studies of the novel PH candidate gene *MOB2 in vivo* and in COs [28] and from *Eml1* mutant HeCo mice [12,61], indicating that this may be a common mechanism underlying the aetiology of PH and other NMDs. Besides, the cilium-localised GTPase ARL13B regulates both cell cycle progression and cell migration [62], connecting the observed changes in proliferation and neuronal migration in proper neurogenesis. Altogether, we identified changes in both the microtubule and actin cytoskeleton and the polarity of aRG. Loss of apically localised cilia and reduction in the stability of the apical F-actin belt suggest aRG delamination and formation of ectopic rosettes upon *Ece2* KD *in vivo*. These ectopic progenitors did not in general lack apico-basal polarity as was visible in the formation of ecopic apical surfaces with Arl13b, but lacked bipolar morphology, possibly leading to a disorientation of

the derived neurons, which may then form nodules as observed after *Ece2* KD in the developing mouse cortex. Hence, our data suggest that *ECE2* directly or indirectly plays a role not only in neuronal migration and in cell fate, but also in cytoskeletal organisation and apico-basal polarity of aRG.

ECE2's role in cytoskeletal integrity was further supported by proteomic analysis: the set of proteins significantly downregulated in *ECE2* KO COs lysate was enriched for microtubule and MAPs, for actin-associated and actin-organising proteins and for guidance and adhesion factors. One downregulated F-actin cross-linker protein was FLNA, mutations in which were identified as genetic cause for familial PH more than 20 years ago [14]. Upon acute *FlnA* KD in the developing rat cortex, the polarised RG structure is disrupted in a similar way to what we detected upon *ECE2* KD and KO and both cell cycle progression and neuronal migration towards the CP are altered [11]. Comparable phenotypes were identified upon cKO of *Mekk4* in mice [63] and after acute *Rcan1* KD in rats [64], both of which act in the FLNA pathway. In the group of guidance and adhesion factors, for example, several SLITs and FLRTs were downregulated in *ECE2* KO COs. This was also found in other COs derived from patients with PH showing similar disruption of the neuroepithelial integrity [13], thus representing another possible common mechanism in PH. In the context of neuronal migration, this is of relevance as differential expression of adhesion molecules such as FLRT1 and FLRT3 between future gyri and sulci was identified to influence the migration of cortical neurons, thereby regulating gyrification independently of progenitor cell expansion [65].

Further, considering the importance of PTMs of cytoskeletal and associated proteins for their function, with phosphorylation as the most widespread modification [66] known to regulate neuronal polarity and morphology [54], we analysed the whole proteome of CTRL and *ECE2* KO COs for differentially phosphorylated protein sites. This analysis revealed an enrichment of actin and microtubule cytoskeleton-related proteins among differentially or exclusively phosphorylated sites, suggesting an additional role of ECE2 in the regulation of the actin and microtubule cytoskeleton through PTMs such as phosphorylation. We thus propose that the misregulation of PTMs can contribute to the aetiology of PH.

## Extracellular function of ECE2: ECM

Whole-proteome analysis of CO lysate unravelled a novel mechanism of action of ECE2: the largest group of proteins downregulated in *ECE2* KO COs compared to control consisted of a total of 49 ECM proteins. Additionally, this is to our knowledge the first study applying MS-based proteomics on CO-conditioned culture medium to analyse differences in their secretome. We identified two significantly downregulated proteins in the secretome of *ECE2* KO COs. These were both collagen-binding proteins, thus supporting the involvement of ECE2 in ECM generation or regulation. Besides, this stresses the strength of COs as *in vitro* models as they generate an elaborate ECM.

Extracellular matrix proteins such as collagens, galectins and lumican have been implicated both in neuronal migration and differentiation and in cortical expansion and folding [67–69], with human-enriched ECM components contributing to interspecies differences such as in gyrification [70]. The collagen VI family, three members of which were downregulated in *ECE2* KO CO lysate, forms microfilament nets in the ECM, where it not only modulates

the stiffness and mechanical properties of the ECM, but also impacts several intracellular processes and pathways [71]. Furthermore, collagens and ECM remodelling are relevant to the onset of epilepsies [72], which patients with PH often present with, thus highlighting a fundamental role of the ECM in brain physiology and pathology. Mutations in *COL6A2* were linked to progressive myoclonus epilepsy [73], and *COL6A1* was reported within a set of genes that are upregulated in the human cerebral cortex with respect to non-human primates, indicating a human-specific role of this gene in brain development and evolution [74].

Here, the identified overrepresentation of ECM components among the downregulated proteins may explain the non-cell-autonomous nature of mechanism observed upon modulation of *ECE2* expression both in *in vitro* and in *in vivo* model systems.

### Similar effects upon KD and OX of *ECE2*

To further analyse ECE2's role in the developing cortex, we acutely overexpressed *ECE2 in vivo* and *in vitro*. Both in the developing mouse brain and in human COs, *ECE2* OX resulted in non-cell-autonomous mislocalisation of neurons in the progenitor zone that strongly resembled that observed upon acute *Ece2/ECE2* KD. Acute OX of *ECE2* in *ECE2* KO COs, on the other hand, resulted in a rescue of neuronal positioning: The number of ectopic neurons relative to the electroporation size was significantly reduced with restored *ECE2* expression. This suggests that the levels of *ECE2* expression during development must be tightly regulated to ensure the correct positioning of neurons and indicates that both increased and decreased levels of ECE2 can equally result in PH-like neuronal mispositioning. At first glance, it may seem surprising that downregulation and forced expression of *ECE2* result essentially in a very similar phenotype, but especially considering the changes that ECE2—through its role in ECM production and regulation—promotes in the niche, we can suggest that local modifications of ECE2 levels can disrupt normal signal distribution and influence neuronal migration and differentiation.

### Summary

In conclusion, we studied the role of *ECE2* in brain development and identified *ECE2* as a promising novel candidate gene for human NMDs as two individuals with biallelic mutations in *ECE2* are affected by PH. We have contributed to unravel the role of *ECE2* in cortical development and suggested changes in the ECM as a possible novel mechanism involved in the complex formation of PH during human brain development. Besides, our findings stress that—although genetically heterogeneous—PH often arises from similar cellular mechanisms.

## Materials and Methods

### Candidate gene prioritisation

The recent trio-based whole-exome sequencing study in patients with PH [27] was interrogated for genes with rare (MAF < 0.005) biallelic variants (homozygous recessive or compound heterozygote) in two or more patients (under the study protocol CEN/11/

12/066 Central Regional Ethics Committee, New Zealand). As outlined in Table EV1 of this report [27], only three genes remained of interest—*ECE2*, *GPR98* and *PTPN13*. Note, although two separate homozygous recessive variants are identified for *SPEG*, they are all present within the same patient. As a further filter to identify a potential candidate gene for functional investigation, we restricted our analysis to any of these three genes whose variants in each individual were reported as potentially deleterious by one of four *in silico* prediction programs [75–78] (see Table EV1). Such prediction programs asses functional (i.e. the resulting properties of the amino changes) and/or evolutionary (such as sequence conservation) properties. Although such genic effects are not a proxy for disease association, any gene satisfying these criteria in this study would be prioritised for detailed functional investigation that further corroborates insight into the potential effect of the variants on gene function. Here, only one gene was prioritised—*ECE2*.

### Reprogramming of fibroblasts to induced pluripotent stem cells

Human-iPSCs were obtained by reprogramming of foreskin fibroblasts as described before [28].

### iPSC culture

Induced pluripotent stem cells were cultured at 37°C, 5% $CO_2$ and ambient oxygen level on plates coated with Matrigel (*Corning*; diluted 1:100 in DMEM/F12 + Glutamax, Thermo Fisher Scientific) in mTeSR1 medium (StemCell Technologies) with daily medium change. For passaging, iPSC colonies were washed with PBS and incubated with StemPro Accutase Cell Dissociation Reagent (#A1110501; Life Technologies) diluted 1:4 in PBS for 3 min. Detached pieces of colonies were collected by washing with DMEM/F12 and 5-min centrifugation at 300 $g$. They were then resuspended in mTeSR1 supplemented with 10 μM Rock inhibitor Y-27632(2HCl) (#72304; StemCell Technologies) for the first day with splitting ratios 1:3–1:6.

### Generation of neural progenitor cells and neurons from iPSCs

Neural progenitors were generated as described previously [31] (Fig EV1A) with some modifications. Embryoid bodies (EBs) were obtained by incubation of feeder-free iPSC colonies with Collagenase Type IV (#7909; StemCell Technologies) for 15 min, followed by washing with DMEM/F12, manual disruption and scraping with a cell lifter (#3008; Corning Life Sciences). Resulting detached pieces of iPSC colonies were plated in suspension in Neural Induction Medium (NIM) consisting of DMEM/F12 + HEPES (#31330095; Life Technologies) with 1× N2 and B27 supplements (without vitamin A, Thermo Fisher) with medium change every other day. EBs were plated on polyornithine/laminin plates after 1 week of suspension culture, and after another week of culture in NIM, neural rosettes were picked and disrupted manually using a P20 tip. Resulting NPCs were cultured in NPM (NIM + 20 ng/ml basic fibroblast growth factor [bFGF/FGF-2], #100-18b-50; Peprotech) and passaged using Accutase (StemCell Technologies) splitting at a maximum ratio of 1:4. NPCs were only used for up to seven passages. For differentiation to neurons, single NPCs were plated at a density of $10^4$ cells/ $cm^2$ on polyornithine/laminin plates and cultured in NPM for 1

more day to reach about 30% cell density. Afterwards, medium was changed to Neuronal Differentiation Medium (NDM; NIM containing 20 ng/ml BDNF [#248-BD; *R&D Systems*] and 20 ng/ml GDNF [#212-GD; *R&D Systems*]) and was differentiated for 20 days with medium change every 5 days.

## Cerebral organoids

Cerebral organoids were generated starting from 9,000 single iPS cells/well as previously described [32] (Fig EV1B) with the following alterations. Organoids were cultured in 10-cm dishes on a DOS-10L orbital shaker at 37°C, 5% $CO_2$ and ambient oxygen level with medium changes twice a week. Organoids were electroporated at 50 days after plating (see Electroporation of cerebral organoids and Fig 1G) and analysed 4 and 7 dpe. For immunostaining, 14-μm sections of COs were prepared using a cryotome. For quality control and selection of ventricles with dorsal identity to include in the analyses, sections were stained for PAX6 and CTIP2 (see example pictures in Fig EV3G and supplementary figure 18 in Ref. [39]). For analysis at 7 dpe, 41 (CTRL) and 28 (*ECE2* KD miR 1) or 53 (*ECE2* KD miR 2) different ventricles in a total of 17 (CTRL) and 25 (*ECE2* KD) organoids from three independent batches were analysed; for *ECE2* OX in control COs, 30 CTRL and 32 *ECE2* OX ventricles in six organoids each from two independent batches were analysed; and for *ECE2* OX in *ECE2* KO COs, 11 CTRL and 16 *ECE2* OX ventricles in three organoids each from one batch were analysed.

## Generation and validation of microRNAs targeting *ECE2/Ece2* and ECE2-OX constructs

MicroRNAs (miRNAs) targeting human or mouse *ECE2* were generated using the BLOCK-iT system from Invitrogen (Thermo Fisher). MicroRNA sequences were determined using Invitrogen's RNAi design tool https://rnaidesigner.thermofisher.com/rnaiexpress/setOption.do?designOption=mirnapid=1961720787891316464. Three microRNA sequences per species were chosen and ordered as oligonucleotides from Sigma. *ECE2*-targeting microRNA oligonucleotides were annealed and ligated into a GFP-containing entry vector pENTR-GW/EmGFP-miR containing the microRNA backbones using T4 DNA Ligase (Thermo Fisher), following the recommendations in the BLOCK-iT system. Subsequently, the microRNA sequences (plus GFP) were cloned into the pCAG-GS destination vector using the Gateway system (Thermo Fisher) according to the manufacturer's recommendations. The resulting microRNA expression plasmids were sequenced using emGFP forward primer (5′-GGCATGGACGAGCTGTACAA-3′).

The knockdown efficiency of the microRNAs was tested via qPCR (Figs EV1E and EV2A) with validation of those targeting human *ECE2* in SH-SY5Y neuroblastoma cells and validation of those targeting mouse *Ece2* in P19 cells. The most efficient constructs were used for neural progenitor transfection in COs by electroporation (see Electroporation of cerebral organoids and Fig 1D) and in the developing mouse cortex by IUE (see *In utero* electroporation). The shown constructs were chosen. A microRNA with scrambled sequence ("miRneg") was used as negative control.

For *ECE2* overexpression in COs and in the developing mouse cortex, pcDNA3.1$^+$/C-(K)-DYK containing human *ECE2* transcript variant 5 was ordered from *GenScript* (#OHu18010). Transcript variant 5 was chosen as it has been identified as specifically abundant in the brain and adrenal gland [79]. For visualisation, this construct was co-electroporated in 2:1 ratio with pCAG-IRES-GFP, using pCAG-IRES-GFP alone as control vector for OX experiments.

## Plasmid preparation

Small-scale plasmid preparation was carried out using the QIAprep Spin Miniprep Kit (Qiagen) and large-scale preparation with the Plasmid Maxi Kit (Qiagen) after transformation in Subcloning Efficiency™ DH5α™ Competent Cells (Thermo Fisher).

## Electroporation of cerebral organoids

For transfection of aRG in germinal zones of COs by electroporation (see scheme in Fig 1D), COs were kept in NDM + A without antibiotics and antimycotics. The organoids were placed in an electroporation chamber (Harvard Apparatus), and miR*ECE2* or miRneg expression plasmid (containing GFP) at a concentration of 1 μg/μl, or *ECE2*-OX plasmid in 2:1 ratio with pCAG-IRES-GFP (total concentration of 1 μg/μl) vs. pCAG-IRES-GFP as control, always supplemented with Fast Green (0.1%; Sigma) for visualisation, was injected into ventricle-like cavities at several positions per organoid. Electroporation was performed with an ECM830 electroporation device (Harvard Apparatus) by subjecting the organoids to a 1-second interval with five pulses of 50-ms duration at 80 mV. Medium was changed to antibiotic-containing NDM + A on the next day.

## *In utero* electroporation

As approved by the Government of Upper Bavaria under the licence number 55.2-1-54-2532-79-2016, pregnant C57BL/six mice were used. They were weighed and anaesthetised by

| Oligo name | Sequence (5′–3′) |
|---|---|
| miR1 *ECE2* top | TGCTGTAATGTTCCAACCACCAATCTGTnTGGCCACTGACTGACAGATTGGTTTGGAACATTA |
| miR1 *ECE2* bottom | CCTGTAATGTTCCAAACCAATCTGTCAGTCAGTGGCCAAAACAGATTGGTGGTTGGAACATTAC |
| miR2 *ECE2* top | TGCTGTTTCCCTGCTCTATCTGCTTTGTnTGGCCACTGACTGACAAAGCAGAGAGCAGGGAAA |
| miR2 *ECE2* bottom | CCTGTTTCCCTGCTCTCTGCTTTGTCAGTCAGTGGCCAAAACAAAGCAGATAGAGCAGGGAAAC |
| miR *Ece2* top | TGCTGTCAATGAGGTCTCTAAGTGGCGTnTGGCCACTGACTGACGCCACTTAGACCTCATTGA |
| miR *Ece2* bottom | CCTGTCAATGAGGTCTAAGTGGCGTCAGTCAGTGGCCAAAACGCCACTTAGACCTCATTGAC |
| miRneg top | TGCTGAAATGTACTGCGCGTGGAGACGTnTGGCCACTGACTGACGTCTCCACGCGCAGTACATTTCAGG |
| miRneg bottom | CCTGAAATGTACTGCGTGGAGACGTCAGTCAGTGGCCAAAACGTCTCCACGCGCAGTACATTTCAGCA |

intraperitoneal injection of saline solution containing fentanyl (0.05 mg per kg body weight), midazolam (5 mg per kg body weight) and medetomidine (0.5 mg per kg body weight; Btm licence number 4518395), and embryos were electroporated at E13 as described [80]. Plasmids were mixed with Fast Green (0.1%; Sigma) and injected at a final concentration of 1 µg/µl. After wound closure and sterilisation, anaesthesia was terminated by injection of buprenorphine (0.1 mg per kg body weight), atipamezole (2.5 mg per kg body weight) and flumazenil (0.5 mg per kg body weight). Brains were fixed 1 or 3 dpe in 4% paraformaldehyde (PFA) for 4 h (1 dpe) or overnight (3 dpe) and were subsequently transferred to 30% (w/v) sucrose in PBS for cryoprotection before embedding and freezing in OCT (VWR). For immunohistochemistry, 12-µm sections were prepared using a cryotome. For each experiment, at least five different mouse brains per condition were analysed.

## Analysis of cerebral organoids

For immunohistochemistry, cerebral organoids at the desired TP were washed with PBS and fixed in 4% PFA for 2 h at 4°C. They were then left in a 30% sucrose solution overnight at 4°C. After this, they were embedded in embedding moulds with OCT (VWR), embedding 3–4 organoids per mould, and kept at −20°C until further processing. Frozen organoid samples were cut at a cryotome, preparing 14-µm sections for immunohistochemistry and 16-µm sections for ISH.

## Immunohistochemistry

Frozen organoid or mouse brain sections were thawn to RT for 20 min and then rehydrated in PBS for 5 min. For nuclear antigens, an antigen retrieval step (HIER) was performed, boiling the sections in 0.01 M citric buffer (pH 6) for 1 min at 720 W and 10 min at 120 W. The slides were then left to cool down for 20 min. Half of the citric buffer was replaced by $H_2O$, and slides were incubated for another 10 min and then washed in PBS. Subsequently, a postfixation step of 10 min was carried out with 4% PFA in PBS. Then, the sections were permeabilised using 0.3% Triton X-100 in PBS for 5 min. After permeabilisation, sections were blocked at RT for at least 1 h with 10% normal goat serum (#VEC-S-100; *Biozol*) and 3% BSA (#A4503-50G; Sigma-Aldrich) in 0.1% Tween in PBS. The primary antibody (see section Primary antibodies) in blocking solution was then incubated overnight at 4°C. Following several washes with 0.1% Tween in PBS, sections were incubated with 1:1,000 dilutions of Alexa Fluor-conjugated secondary antibodies (Life Technologies) in blocking solution for at least 1 h at RT, using 0.1 µg/ml 4,6-diamidino-2-phenylindole (DAPI, Sigma-Aldrich) to counterstain nuclei. Finally, sections were washed again several times with 0.1% Tween in PBS and mounted with Aqua PolyMount (#18606; Polysciences). Sections were visualised using a Leica SP8 confocal laser scanning microscope. For immunohistochemistry of NPCs and neurons, cells were cultured on round coverslips (13 mm diameter, VWR) in 24-well plates, washed with PBS and fixed with 4% PFA in PBS for 15 min at RT. For HIER, citric buffer was boiled, added to coverslips and left to cool down, while permeabilisation, blocking and staining were carried out as described for the tissue sections.

## Primary antibodies

| Antigen | Dilution | Vendor | Catalogue # |
|---|---|---|---|
| ARL13B | 1:500 | Proteintech | 17711-1-AP |
| ACETYLATED TUBULIN | 1:6,000 | Sigma-Aldrich | *T7451* |
| BLBP | 1:150 | Merck Millipore | ABN14 |
| CTNNB1 (β-catenin) | 1:500 | BD Biosciences | 610154 |
| CTIP2 | 1:500 | Abcam | ab18465 |
| Doublecortin (DCX) | 1:2,000 | Merck Millipore | ab2253 |
| ECE2 | 1:300 | Novus Biologicals | NBP1-81495 |
| F-actin (Phalloidin-594) | 1:40 | Thermo Fisher | A12381 |
| GAPDH | 1:6,000 | Merck Millipore | CB1001 |
| GFP | 1:1,000 | Aves Lab | GFP-1020 |
| MAP2 | 1:500 | Sigma-Aldrich | M4403 |
| NANOG (clone 7F7.1) | 1:250 | Merck Millipore | MABD24 |
| NEUN | 1:500 | Merck Millipore | MAB377 |
| OCT4 | 1:500 | Merck Millipore | ABD116 |
| PAX6 | 1:500 | Biolegend | PRB-278p |
| SOX2 | 1:500 | Cell Signalling | 2748S |
| Phospho-Histone H3 (PH3) | 1:500 | Millipore | 06-570 |
| P-Vimentin (Ser55) | | Biozol | D076-3 |
| SATB2 | 1:500 | Abcam | ab51502 |
| SSEA1 (eBioMC-480 [MC-480]), Alexa Fluor 488 | 1:200 | eBioscience™ | 53-8813-41 |
| TRA-1-60 | 1:250 | Merck Millipore | MAB4360 |
| TUBB3 | 1:500 | Sigma-Aldrich | T8660 |

## Generation of isogenic *ECE2* Exon3/4 KO iPSC line

Two CrRNAs (one each in exon 3 and exon 4) of the *ECE2* gene were selected based on Geneious Prime, CCtop [81] and the guide design tool by Zhang Lab (CRISPR.MIT.EDU). The selected exon 3-CrRNA (5′ GTTAGGCTCACGCACGCAGC 3′) and exon 4-CrRNA (5′ GTTCCAGCGAGAACGCCCAT 3′) were individually assembled and *in vitro*-transcribed into guide RNA (gRNA) using the EnGen sgRNA Synthesis Kit, *S. pyogenes* (#E3322S; *NEB*). The *in vitro*-transcribed guides were assembled into a gRNA-CAS9-NLS RNP complex by incubating the CAS9-NLS (#M0646M; NEB) and both *in vitro*-transcribed gRNAs for 20 min at RT, followed by electroporation into single cells of the control iPSC line (see above) using Amaxa 4D nucleofector (Lonza). Electroporated cells were plated onto a 10-cm Geltrex (#A1413302; Thermo Fisher)-coated dish supplemented with mTesR1 (#85850; StemCell Technologies) and Y-27632 (#72308; StemCell Technologies). The electroporated cells were allowed to recover for 10 days with daily media changes of mTeSR1 before single-cell live sorting into Geltrex-coated 96-well plates using CloneR (#05888; StemCell Technologies). The iPSC colonies were screened for INDELS using the protocol by Yusa K [90] with the following primers:

FW: 5′-CAGACTCAAGGCTCAACTCACTGGCTGGCCTCATTG-3′
RV: 5′-CCATCAAACCCATCTGCACTTCTCAATAGTCC-3′

Putative positive clones were grown as individual KO lines. Genomic DNA was isolated from the generated KO iPSC lines using QIAamp DNA Mini Kit (#51306; Qiagen) and used as template with the above-mentioned primer pair. The resulting PCR product (906 bp) was cloned using NEB PCR Cloning Kit (#E1202S; NEB) and Sanger-sequenced to screen the individual genomic alleles. All the Sanger sequencing was performed at the Sequencing Facility of the Max Planck Institute of Molecular Cell Biology and Genetics, Dresden, Germany. Standard G-banded karyotyping analysis was performed and showed no chromosomal abnormalities in the *ECE2* KO iPSC lines. Pluripotency was confirmed by IHC using TRA1-60, OCT4 and NANOG.

Manufacturer's recommendations and kit protocols were followed for all the kits and reagents mentioned above.

Two *ECE2* KO clones with the following modifications resulted: Allele 1 with a 32-bp deletion causing a frameshift in exon 3, followed by a premature STOP codon; and Allele 2 with a 463-bp deletion causing the fusion of first half of exon 3 with the second half of exon 4 at DNA levels, resulting in a deletion of 80 amino acids between exons 3 and 4, including the patient mutation.

### RNA isolation and RT–qPCR for *ECE2*/Ece2 levels in microRNA validation and in NPCs and neurons

Cells were washed and scraped with cold PBS, and the pellet was lysed in QIAzol® Lysis Reagent (#79306; Qiagen). NPCs and neuronal identities were confirmed by qPCR for PAX6, NESTIN, DCX and NEUN. RNA was extracted using RNA Clean & Concentrator Kit (#R1015; Zymo Research), and cDNA was synthesised from 2 μg RNA each using SuperScript III Reverse Transcriptase (#18080-044; Thermo Fisher) with oligo(dT)$_{12-18}$ primers (#18418012; Thermo Fisher) according to the manufacturer's protocol. Subsequently, qPCR was performed in triplicates on a LightCycler® 480 II (Roche) using the LightCycler® 480 SYBR Green I Master (#04707516001; Roche) with the following reaction mix 1 μl of each primer (5 μM), SYBR Green I Master 5 μl, H$_2$O 2 μl and 1 μl of cDNA. The primer sequences were as follows:

for *ECE2*:
FW: 5′-AGAAGTCCTGTGTGCCGAGG-3′
RV: 5′-ATGCGGTCCGGATTTCGCT-3′
for *Ece2*:
FW: 5′-AGTGTTGGCTGCCCTTCTTTT-3′
RV: 5′-CAGCTACTCGAATGCAGGCTT-3′
for *GAPDH*:
FW: 5′-AATCCCATCACCATCTTCCAGGA-3′
RV: 5′-TGGACTCCACGACGTACTCAG-3′

qPCR was done with: initial denaturation at 95°C for 10 min, then 45 cycles of 95°C for 10 s, 60°C for 10 s and 72°C for 10 s, and a melting curve from 95 to 65°C with 15 s per step. $C_p$ values were determined from the second derivative maximum of the amplification curve, and fold expression was calculated using the $\Delta\Delta C_p$ method.

### In situ hybridisation on cerebral organoid sections

*In situ* hybridisation was performed based on exons 7–12 of full-length *ECE2* coned into pBluescript SK(+) (ordered from GenScript).

For the generation of probes, 20 μg of plasmid was digested for linearisation with EcoRV for the sense and Kpn1 for the antisense probe (NEB). DNA was extracted using phenol–chloroform–isoamyl alcohol, pelleted using ammonium acetate and resuspended in RNase-free water. *In vitro* transcription was performed from 1 μg each of the linearised plasmids using T7 polymerase for the sense probe and T3 for the antisense probe, together with the digoxigenin (DIG) RNA labelling mix (Roche) according to the manufacturer's protocol.

For detection of mRNA transcripts, 16-μm frozen sections were hybridised with 1 μl probe in 150 μl hybridisation buffer (formamide 50%, 1× SSC buffer [20× stock: 3 M sodium chloride, 300 mM trisodium citrate, adjusted to pH 7.0 with HCl], 2% SDS, 50 μg/ml yeast tRNA [Roche], 2% blocking reagent [Roche] 2% and 50 μg /ml heparin [Sigma]) per slide at 65°C overnight.

The next day, hybridisation buffer was replaced by washing buffer (50% formamide, 1× SSC and 0.1% Tween-20 in H$_2$0) and the sections were washed at 65°C for 10 min. This step was repeated twice for 30 min. Subsequently, the washing buffer was replaced by MABT (5× stock: 100 mM maleic acid, 150 mM sodium chloride and 0.1% Tween-20 in H$_2$O, pH adjusted to 7.5 with sodium hydroxide). Sections were incubated in MABT twice for 30 min at RT. Subsequently, sections were blocked using ISH blocking buffer (1× MABT, 2% blocking reagent [Roche] and 2% FCS in H$_2$O) for 1 h at RT. Anti-DIG Fab fragments (Roche) were then diluted 1:2,500 in ISH blocking buffer and incubated on the sections at 4°C overnight.

The next day, sections were washed with MABT solution five times for 20 min. Afterwards, sections were incubated twice for 10 min in AP staining solution (100 mM Tris, pH 9.5, 50 mM magnesium chloride, 200 mM sodium chloride and 0.1% Tween-20 in H$_2$O). The AP staining solution was then replaced by AP staining solution containing NBT and BCIP (both Roche) in a 0.35% concentration, using 150 μl per slide. The colour reaction was allowed to proceed overnight at 4°C and stopped by washing the sections with PBS. Sections were then stained with DAPI (0.02 mg/ml in PBS) to visualise nuclei.

### Drug treatment and 2D time-lapse imaging of young neurons

For time-lapse imaging, NPCs were plated on a polyornithine/laminin-coated 24-well plate and left in neural progenitor medium (DMEM/F12 with HEPES and glutamine, N2 1:100, B27 without vitamin A 1:50) without bFGF for 4 days with medium changes every other day to allow the cells to start differentiation. The control condition (CTRL) was additionally treated with DMSO 1:1,000, whereas in the other wells, 1 μM Phosphoramidon (PHOS; #R7385; Sigma) was added for ECE2 enzymatic inhibition [42]. After this, cells were imaged for 17 h using a Zeiss Axiovision Observer fluorescence microscope (Zeiss), maintaining 37°C and 5% carbon dioxide concentration. The cells were imaged every 10 min, several positions being taken per well. From the time-lapse movies, movement data of single cells were derived using the Manual Tracking plugin for ImageJ [82]. No centring correction was applied. From the tracking data, cell movement parameters were calculated using Microsoft Excel 2013. Neuronal identity of tracked cells was confirmed after the time-lapse imaging by TUBB3 immunostaining.

The average velocity was directly calculated from the measured velocities. The number of resting TP was calculated by counting the number of time points with a velocity of 0, and dividing this number by the length of the whole dataset for each cell. The tortuosity was calculated as division of the length of the path actually migrated by the net displacement of a cell from the beginning to the end of the track. Data are shown as z-scores calculated relative to the mean and standard deviation of the control within each experiment ($N$ = total number of cells from two independent experiments). Statistical significance was tested via 2-tailed chi-square test.

### Image acquisition

All imaging except for time-lapse microscopy was performed using a Leica SP8 with the LASX Software from Leica and 25× and 40× water immersion objectives. Images were acquired in sequential xyz mode with at least 1,024 × 1,024 pixels and 1–1.5 μm thickness of optical z layers. Images were saved as .lif file format to contain the metadata for the subsequent image analysis.

### Image analysis

Image analysis was done in Fiji [82] using the manual cell counter and length and area measurement tools, as well as the scale bar tool. Images are shown as Z-stacks. For the quantification of ectopic neurons upon *ECE2* KD or OX, all NEUN$^+$ cells in the progenitor zone within the electroporated area were counted. For normalisation to the electroporation size, this number was related to the number of GFP$^+$ RG. For the quantification of PH3$^+$ mitotic NPCs relative to (electroporated) apical surface as done in *ECE2* KO COs and upon *ECE2* OX in CTRL COs, the total number of PH3$^+$ cells in the respective (electroporated area of the) germinal zone was counted and divided by the (electroporated) apical surface length of the VZ'.

The size of cerebral organoids was measured based on the perimeter from brightfield images acquired at a Zeiss Axiovision Observer fluorescence microscope (Zeiss) with at least 10 organoids per TP and batch.

Cell quantifications in the developing mouse cortex 1 dpe were done on a single image of the z-stack with the brightest staining in the middle of the stack by binning of the electroporated area, from apical to basal surface, into five equally big areas (Fig EV2B). Afterwards, single cells were counted using the manual cell counter. For quantification of brains at 3 dpe, 3–8 brains per condition were analysed concerning the presence of ectopically positioned progenitors or deep- or upper layer neurons (at least five cells per parameter) and concerning the quality of the apical belt, including the appearance of additional ectopic apical surface in a yes-or-no system.

### Quantification of intruding MAP2

To assess zonal disorganisation in COs upon *ECE2* KO, the amount of ventricles in CTRL and ECE2 KO organoids with intruding MAP2$^+$ neuronal processes and/or with MAP2$^+$ cell bodies localised in the progenitor zone was counted. In detail, VZs containing more than three MAP2$^+$ processes were categorised as "intruding processes" and those with MAP2$^+$ cell bodies fell into the second, even more disorganised category.

### Measurement of the thickness of apical F-actin

The thickness of the apical F-actin belt in ventricle-like zones of COs was determined by drawing and measuring the area of the Phalloidin-594-positive belt, measuring the length of the apical belt in its middle in parallel to the apical ventricular surface and dividing the area by the length.

### Quantification of acetylated tubulin

The quality of the microtubule cytoskeleton in COs was assessed by IHC for acetylated tubulin as marker for stabilised microtubules. In independent germinal zones of COs, the intensity of the acetylated tubulin signal was measured in Fiji as mean grey value in a region of interest including both progenitor and neuronal layers.

### Quantification of Arl13B

Apical ARL13B in CO ventricles was rated as "1" for high signal and density of primary cilia and as "0" in case of reduced signal intensity and/or reduced apical localisation of primary cilia.

### FACS analysis

For every FACS sample, three cerebral organoids were pooled at 60 days and two organoids were pooled at 107 days. Organoids were washed with PBS and then enzymatically dissociated to single cells using Accutase (StemCell Technologies) for 3 × 10 min at 37°C with gentle trituration using a P1000 pipette in between. After dissociation, samples were washed with PBS and centrifuged at 300 $g$ for 5 min. Subsequently, the suspension was filtered through a 100-μm cell strainer and cells were collected by centrifugation at 300 $g$ for 5 min. Cells were then fixed with 5 ml ice-cold 70% EtOH for 1 h at −20°C. Afterwards, they were centrifuged for 30 min at 376 $g$ and 4°C, resuspended in 5 ml staining buffer (1% FBS in PBS) for washing and centrifuged by further centrifugation for 30 min at 376 $g$ and 4°C, after removing all except for 50 μl supernatant containing the cells that were kept at 4°C. A few microliter of each sample was collected and pooled to be used as isotype controls (ICs). Staining was performed the next day by diluting primary antibodies in staining buffer (concentrations different from Antibodies Table: gp-α-DCX 1:1,000; mIgG1-α-MAP2 1:500; rb-α-PAX6 1:300; rb-α-PH3 1:300) and adding 20 μl resulting staining solution to each sample and incubating on ice for 30 min, followed by addition of 1 ml staining buffer each and 30-min centrifugation for washing. Supernatant was again removed, leaving the cells in 50 μl leftover staining buffer. They were resuspended with staining solution containing the Alexa Fluor-conjugated secondary antibodies (1:800 final dilution), incubated for another 30 min on ice and washed as before. ICs were also immunostained with secondary antibodies. The labelled cells were then resuspended in PBS for FACS analysis. FACS analysis was performed using a FACS Aria (BD) in BD FACSFlow™ medium with a nozzle diameter of 100 μm and a flow rate below 500 events/sec. Debris and cell aggregates were gated out by forward scatter (FSC)–side scatter (SSC); single cells were gated out by FSC-W/FSC-A. Gating strategies for

fluorophores were selected using ICs for the different secondary antibodies. Examples of the gating strategy are shown in Fig EV3C and G.

## Statistics

Statistics for quantifications of immunohistochemical stainings in COs and embryonic mouse brain was performed in SigmaPlot (version 13.0; Systat Software, San Jose, CA) unless stated otherwise below. Data were tested for normal distribution by the Shapiro–Wilk test. Normally distributed data were analysed further by the Brown–Forsythe test for equal variance. One-way ANOVA and Tukey's test were then applied for normally distributed data of equal variance. The Kruskal–Wallis ANOVA on ranks and Dunn's pairwise multiple comparison were used for non-normally distributed data. Exact binomial test was done using GraphPad Prism. The Mann–Whitney $U$-test and hierarchical clustering analysis using chi-square test were done in STATISTICA (StatSoft Inc.). For time-lapse imaging results, statistical significance was tested in SPSS (Statistical Package for Social Science, IBM®) via two-tailed chi-square test.

For electroporation of microRNAs targeting *ECE2* in COs, four batches of organoids were analysed for each construct with several organoids each. Data are shown with $n = t$otal number of electroporated ventricles analysed per construct from organoids from all batches combined. For *ECE2* OX in COs, two batches of organoids were analysed with several organoids each and the number of analysed ventricles was shown as $n$.

For *in vivo* analyses after IUE, three CTRL and four *Ece2* KD brains were analysed with at least two sections per brain in the binning analysis at 1 dpe (E13–14) and six brains per condition at 3 dpe (E13–16); for *ECE2* OX *in utero*, three CTRL and five *ECE2* OX brains were analysed 1 dpe and four CTRL and eight *ECE2* OX brains 3 dpe.

For 60-day-old *ECE2* KO COs, FACS analysis, WB and MS were performed on 3–4 independent batches and IHC was performed on COs from 2 different batches. In detail, three organoids were combined to be one FACS or WB sample (1 $n$), 20 organoids per condition and batch were combined for MS, and IHC was done using six organoids per condition and batch. For 107-day-old *ECE2* KO and CTRL COs, organoids from one batch were analysed, each sample ($n$) containing two pooled organoids.

Whenever data are shown as box plot, the mean is marked by a red line, and the median, by a black line, and the box represents 25th and 75th percentiles with whiskers extending to 10th and 90th percentiles, with all outliers shown.

## G-actin vs. F-actin fractionation by ultracentrifugation

Per sample, three COs were pooled. The medium was removed, and they were washed once with PBS. Then, the COs were homogenised in 500 µl actin lysis buffer (50 mM NaCl, 1 mM EDTA, 0.5% Triton X-100, 20 mM HEPES, pH 7.9, with 1× protease inhibitor mix added just before use; pre-warmed to 37°C) by pipetting up and down and soaking the lysate three times through a 0.4-mm-diameter syringe. To further homogenise, samples were incubated at 37°C for 10 min. Afterwards, pellet and supernatant fractions were prepared by ultracentrifugation at 100,000 $g$ for 1 h at

4°C. Supernatants, containing G-actin, were collected and mixed directly with SDS–PAGE loading buffer. To pellets, containing F-actin, another 500 µl actin lysis buffer each was added and they were sonicated with a Bioruptor Pico (Diagenode) applying five cycles with 30 s ON and 30 s OFF each. Then, SDS–PAGE loading buffer was also added to the F-actin fraction and both G- and F-actin fractions were boiled for 5 min at 95°C prior to separation by Western blot on 12% SDS gels.

## Western blot

Cells or whole cerebral organoids were collected and washed with PBS prior to lysis in lysis buffer (62.5 mM Tris–HCl, pH 6.8, 2% SDS and 10% saccharose in $H_2O$ with protease inhibitor mix [Roche] added just before use), and protein concentration was determined with the Pierce™ BCA Protein Assay Kit (#23227; Thermo Fisher). Twenty-microgram protein per lane was then loaded and separated by SDS–PAGE with 10–12% gels. Proteins were transferred onto a nitrocellulose membrane (GE Healthcare) with blotting buffer (129 mM glycine, 25 mM Tris-base and methanol 20%) for 90 min at 100 V. After blotting, the membrane was blocked with 5% milk powder in TBS-T before incubation with the primary antibody in 2.5% milk in TBS-T at 4°C overnight (see section Primary antibodies). After washes with TBS-T, membranes were incubated with HRP-conjugated secondary antibody diluted 1:10,000 in 2.5% milk in TBS-T for 1 h at RT. For protein detection, membranes were incubated with ECL detection reagent (Millipore) and visualised using the ChemiDoc System (Bio-Rad). Band intensities were quantified using the Image Lab™ Software (Bio-Rad).

## Whole-mount staining and clearing (iDISCO)

For staining and clearing of whole organoids, the iDISCO method was used [45] with the antibodies staining ARL13B, NEUN and DCX and DAPI for nuclear counterstain. Prior to use in whole-mount staining, antibodies were tested for methanol (MeOH) compatibility as suggested. The staining and clearing protocol was then followed with minor changes. In detail, organoids were dehydrated with increasing volumes of MeOH in water (20, 40, 60, 80 and 100% MeOH), 1-h incubation at RT each, while shaking. Organoids were washed further with 100% MeOH and incubated for 1 h at 4°C. Then, organoids were incubated with a 66% dichloromethane (DCM)/33% MeOH solution overnight at RT in continuous agitation. The following day, they were washed twice in 100% MeOH for 15 min at RT. Samples were subsequently bleached in 5% $H_2O_2$ in MeOH overnight at 4°C. The samples were then rehydrated with decreasing volumes of MeOH in water (80, 60, 40, 20% and PBS), continuously shaking for 1 h each at RT. After this, the samples were permeabilised using a 0.2% Triton X-100 solution in PBS twice for 1 h at RT. In order to further facilitate the permeabilisation of primary antibodies in following steps, the samples were incubated in permeabilisation solution (0.2% Triton X-100 in PBS + 2% DMSO, 0.3 M glycine) at 37°C for 1 day. Following permeabilisation, samples were incubated with 0.2% Triton X-100 in PBS + 10% DMSO and 6% normal goat serum at 37°C for 1 day for blocking. Primary antibodies (rb-α-ARL13B 1:200, mIgG1-α-NEUN 1:200 and gp-α-DCX 1:800) were diluted in washing

solution (0.2% Tween-20 in PBS, 10 mg/ml heparin) supplemented with 5% DMSO and 3% NGS and added to the samples, incubating at 37°C for 2 days. Afterwards, samples were washed with washing solution five times over the course of 1 day. The Alexa Fluor-conjugated secondary antibodies were diluted 1:300 (DAPI 1:1,000) in washing solution, adding 3% NGS without DMSO, and incubated for 2 days at 37°C. Organoids were then washed 4–5 times over 1 day before clearing.

For clearing, organoids were first dehydrated with increasing concentrations of MeOH in water (20, 40, 60, 80 and 100%) for 1 h each at RT with continuous agitation. The 100% MeOH was repeated twice, and the organoids were kept overnight at this stage. The following day, they were transferred to a solution containing 66% DCM/33% MeOH at RT in continuous agitation. Organoids were washed twice with 100% DCM for 15 min in order to completely remove the MeOH. Eventually, the solution was removed and samples were placed in 100% dibenzyl ether (DBE) without agitation for clearing. Samples were covered completely with the DBE solution to prevent oxidation.

Organoids were positioned in self-made silicon chambers filled with DBE for imaging at the Leica SP8 laser scanning microscope. Videos were generated using Imaris software.

## Whole-proteome analysis from COs

Cerebral organoids were washed twice with PBS and homogenised in ice-cold lysis buffer (1% (v/v) NP-40, 1% (w/v) sodium deoxycholate and 1 tablet protease inhibitor [cOmplete™, Mini, EDTA-free protease inhibitor cocktail, Roche] in 10 ml PBS) using ultra-sonication at 40% intensity for 10 s. Lysis was done for 30 min at 4°C while rotating. The insoluble fraction was pelletised (10 min, 14,000 $g$, 4°C), and protein concentration was determined by bicinchoninic acid assay (BCA, Carl Roth GmbH + Co.). For the next analysis, 200 µg of the total protein was used. Proteins were precipitated by addition of 900 µl acetone and incubated overnight at −20°C. The protein pellet was harvested by centrifugation at 9,000 $g$ for 15 min at 4°C and washed once with 0.5 ml of ice-cold methanol. Proteins were reconstituted in 200 µl digestion buffer (20 mM HEPES, pH 7.5, 7 M urea, 2 M thiourea), reduced (0.2 µl 1 M DTT, 45 min, 25°C) and alkylated (2 µl, 30 min, 25°C, in the dark). The alkylation reaction was quenched by addition of 0.8 µl 1M DTT and incubation for 30 min at 25°C. Proteins were pre-digested with 1 µl LysC (Wako) at 25°C for 4 h. Six hundred microliter 50 mM TEAB buffer was added, and the proteins were digested overnight with 1.5 µl sequencing grade trypsin (0.5 mg/ml, Promega) at 37°C. The following day, the samples were acidified with 10 µl formic acid to a pH of 2–3. Peptides were desalted on 50 mg SepPak C18 cartridges (Waters Corp.) on a vacuum manifold. The cartridges were equilibrated with 1 ml acetonitrile, 1 ml 80% acetonitrile and 3 ml 0.5% formic acid. The samples were loaded on the cartridges and subsequently washed with 5 ml 0.5% formic acid. The peptides were eluted with two times 250 µl 80% acetonitrile and 0.5% formic acid. Samples were combined and dried by lyophilisation. Peptides were reconstituted in 30 µl 1% (v/v) formic acid, prepared for mass spectrometry by filtering through a membrane filter (Ultrafree-MC and Ultrafree-LC, Durapore PVDF—0.22 µm, Merck Millipore) and transferred into mass vials.

## Secretome analysis from COs

For secretome analysis, 5 ml culture medium (NDM + A) was collected from 60-day-old COs after 4 days of conditioning. The medium was cleared from dead cells and debris by centrifugation at 34 $g$ for 5 min. To the cleared medium, 30 ml MeOH was added for protein precipitation at −20°C overnight. Precipitated protein was then harvested by centrifugation at 9,000 $g$ and 4°C for 15 min. Secretome samples were then further prepared for proteomic analysis as described above for the whole CO proteome.

## Mass spectrometry

Mass spectrometry analysis was performed on Q Exactive Plus instrument coupled to an Ultimate3000 Nano-HPLC via an electro-spray easy source (all Thermo Fisher Scientific). Samples were loaded on a 2-cm PepMap RSLC C18 trap column (particles 3 µm, 100A, inner diameter 75 µm, Thermo Fisher Scientific) with 0.1% TFA and separated on a 50-cm PepMap RSLC C18 column (particles 2 µm, 100A, inner diameter 75 µm, Thermo Fisher Scientific) constantly heated at 50°C. The gradient was run from 5–32% acetonitrile and 0.1% formic acid during a 152-min method (7 min to 5%, 105 min to 22%, 10 min to 32%, 10 min to 90%, 10-min wash at 90% and 10-min equilibration at 5%) at a flow rate of 300 nl/min. For measurements of chemical–proteomic samples on the Q Exactive Plus instrument, survey scans ($m/z$ 300–1,500) were acquired in the orbitrap with a resolution of 70,000 at $m/z$ 200 and the maximum injection time was set to 80 ms (target value 3e6). Data-dependent HCD fragmentation scans of the 12 most intense ions of the survey scans were acquired in the orbitrap at a resolution of 17,500, maximum injection time of 50 ms and minimum and maximum AGC targets of 5e3 and 5e4, respectively. The isolation window was set to 1.6 $m/z$. Unassigned and singly charged ions were excluded for measurement, and the dynamic exclusion of peptides enabled for 60 s. The lock-mass ion 445.12002 from ambient air was used for real-time mass calibration on the Q Exactive Plus. Data were acquired using Xcalibur software version 3.1sp3 (*Thermo Fisher Scientific*).

Raw files were analysed using MaxQuant software (1.6.2.10). Searches were performed against the UniProt database for Homo sapiens (taxon identifier: 9606, 21 December 2018, including isoforms). At least two unique peptides were required for protein identification. False discovery rate determination was carried out using a decoy database, and thresholds were set to 1% FDR both at peptide–spectrum match and at protein levels.

Phosphosites were identified by MaxQuant default settings for pSTY and quantified by LFQ. Average protein occupancy of the phosphosites was compared between CTRL and KO COs (Datasets EV4–EV6). Phosphosites identified only in CTRL or KO COs (Dataset EV5) were required to be found at least twice within the replicates with localisation probability higher than 70%.

Statistical analysis of the MaxQuant result table protein-Groups.txt was done with Perseus 1.5.1.6. Putative contaminants and reverse hits were removed. LFQ intensities were log2-transformed, hits with less than three valid values in each group were removed and $-\log_{10}$($P$-values) were obtained by a two-sided one-sample Student's $t$-test over replicates with the initial significance level of $P = 0.05$ adjustment by the multiple testing correction

method of Benjamini and Hochberg (FDR = 0.05), the $-\log_{10}$ of *P*-values were plotted by volcano plot function. Final volcano plots were edited in Origin.

### GO term analysis

Protein interaction network analysis was performed using STRING.db [52]. For GO term analysis of proteins significantly downregulated in *ECE2* KO COs underlying the categories in Fig 6B (see also Table EV2), both STRING.db [52] and PANTHER (panthe rdb.org; [51]), choosing *homo sapiens* as species, were used to identify overrepresented functional protein groups and single proteins falling into those. For the category "neurodevelopmental disorder", the effect of known mutations in each gene was analysed using DECIPHER [50] and focusing on SNVs and small indels (monogenic variants). Dysregulated phosphosites from Dataset EV4, illustrated in Fig 6D, were uploaded into PANTHER database and searched for enrichment of cellular component complete using Fisher's exact test and FDR threshold of 0.05.

## Data availability

The mass spectrometry proteomics data have been deposited to the ProteomeXchange Consortium via the PRIDE repository [83] (https://www.ebi.ac.uk/pride) with the dataset identifier PXD015571.

**Expanded View** for this article is available online.

### Acknowledgements
Thanks to the families for their participation. We thank all members of the laboratory for fruitful discussions, Mariana Schröder and Mira Jakovcevski for help with statistical analysis, and Timucin Öztürk, Vincenza Sportelli and Bianca Schmid for technical support. Thanks to Magdalena Götz for kindly hosting I.Y.B in her lab. This work was supported by funding the Max Planck Society (S.C.-I.Y.B.-G.G.-R.D.G.-C.K.), the Health Research Council of New Zealand and Curekids (A.O.N-S.P.R.), the Graduate School of Systemic Neurosciences (I.Y.B.), European Research Council (ERC) consolidator grant and CIPSM (S.A.S. -P.K.) Alexander von Humboldt fellowship (P.K.), and ERA-Net E-Rare (S.C-C.K.).

### Author contributions
Conceptualisation: IYB, SC; Methodology: IYB, PK, GG, RDG, CK, ACO; Investigation: CK, ACO, RDG, SK; Resources: SPR, SAS, SC; Writing—original draft: IYB; Writing—review & editing: IYB, ACO, SPR, SC; Supervision: SC; Funding acquisition: SPR, SC.

### Conflict of interest
The authors declare that they have no conflict of interest.

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
