## [Review Process File · EMBO Reports]

ECE2 regulates Neurogenesis and Neuronal Migration during Human Cortical Development

Isabel Y. Buchsbaum, Pavel Kielkowski, Grazia Giorgio, Adam C. O'Neill, Rossella Di Giaimo, Christina Kyrousi, Shahryar Khattak, Stephan A. Sieber, Stephen P. Robertson and Silvia Cappello

Review timeline:

Submission date:	1 April 2019
Editorial Decision:	7 May 2019
Revision received:	18 September 2019
Editorial Decision:	20 December 2019
Revision received:	18 February 2020
Accepted:	21 February 2020

Editor: Esther Schnapp

Transaction Report:

1st Editorial Decision

7 May 2019

Thank you for your patience while your manuscript was peer-reviewed at EMBO reports. We have now received the full set of referee comments that I paste below. Honestly, this is a difficult decision, as it seems that none of the referees is convinced that the data are sufficiently strong for publication, and together they ask for quite significant revisions.

I have asked all referees for cross-comments but have not heard back from them. I am also currently at a conference and therefore think that the best way forward is to send you all referee comments now so that you can let me know what you think.

If you think that you can address all concerns, I am happy to invite you to revise the study. If not, we can discuss the specific points further, and I could also consult with the referees again. Does this sound good?

REFeree REPORTS

Referee #1:

Isabel et al. investigate the role of endothelin-converting-enzyme-2 (ECE2) on a neuronal migration disorder, periventricular heterotopia (PH). Previous studies showed the genetic heterogeneity in PH. The authors examined the role of ECE2 in developing cortex by using mouse model and human cerebral organoids. The authors showed that knockdown (KD) ECE2 in cortical organoids led to the accumulation of neurons in the ventricular zone. Furthermore, the KD of ECE2 on developing mouse cortex resulted in more ectopic rosettes and nodules. Moreover, ECE2 inhibition or knock out resulted in changes in the cytoskeleton and apico-basal polarity. Lastly, the proteomics analysis in organoids deciphered the role of ECE2 in extracellular matrix production and integrity. The manuscript presents multiple comprehensive data to describe the function of ECE2 in developing cortex. Despite the interesting findings, following comments need addressed to improve the

manuscript.

1. The authors presented two PH patients containing mutations at ECE2 and the MRI result. However, the authors examined effects of ECE2 by generating organoids with ECE2 KD, or KO, as well as KD in mouse brain, based on the hypothesis that the ECE2 mutations found in patients are loss of function. Authors should discuss this more, or generate iPSC or ESC with the given point mutations to investigate the function of the patients' mutation.

2. In Figure 1C, the expression profile of ECE2 seems an interesting. According the website referenced in the manuscript, some data (ID in the web) for ECE2 show decrease during neural differentiation, and some data show increase during neural differentiation. Authors should describe what each dot represents and provide more representative data. Since the data was not generated by the authors, they may consider move it to supplementary data. Authors may perform qPCR for ECE2, as well as other neuronal, pluripotency, and three germ layer markers during the neural differentiation in 2D and 3D, and present that their neural differentiation was good and expression of ECE2 shows the pattern in Figure 1C.

3. When analyzing the in vivo function of Ece2 in mouse cortex, sections were made as Bin. Authors should show the defining of each bin.

4. In Figure 4A, and 4C, it is difficult to see what the F-actin area represents. Figures for control A and C are very different, although they should represent similar apical adherens belt. Authors need to show the diagram of whole organoid, and mark where the given figures represent. Same is true for 4E, and most of the figures given.

5. In Figure 4F, G, H, and I, are the acetylated-tubulin staining is showing that all cells in ECE2 KO and Phos treated organoids showing lower level of acetyl-Tubulin, or only neurons or aRG are showing the lower? It is difficult to see. Authors may perform the staining of Acetyl-Tubulin in the setting of experiments of Figure 1 and Figure 2 where the KD cells vs control can be distinguished by the presence of GFP. Additionally, authors need to perform Acetyl-Tubulin Western blot for quantification.

In addition, as for the quantification of the phenotype, authors used 1 or 0, as either high or reduced signal or density. This kind of quantification seems very arbitrary and subjective. authors need to used more objective way to quantify.

6. Figure 4J, and L are areas that supposedly represent the ventricular zone containing aRG, but they do not seem to ventricular zone. Again, authors need to show where the figure represent by using a diagram of organoid. Additionally, authors should present more data showing the similar staining. The so-called representative images of the Figures do not seem to represent the average structures of the organoids.

7. Western blot for Figure EV4A is not convincing. It is difficult to see the difference in band intensity of CTNNB1. Additionally, authors should present the total level of ERM in Figure EV4D.

Referee #2:

In this manuscript, Buchsbaum et al investigated the role of ECE2, a potential candidate gene of Periventricular Heterotopia (PH), in neurogenesis during cortical development in mouse and human iPSCs derived cerebral organoids. The main findings of this study are: 1) ECE2 is identified as a PH candidate gene and enriched in neurons and apical surface in cerebral organoids (CO). 2) ECE2 knockdown and inhibition of ECE2 activity cause defects in neuronal migration in COs. 3) ECE2 knockdown leads to ectopic rosettes and nodules during embryonic cortical development in mouse. 4) Alteration in ECE2 activity and expression leads to defects in cytoskeleton in COs. 5) Proteome analyses confirmed the role of ECE2 in cytoskeleton dynamics, and identified its novel role in ECM composition.

These findings on the functional role of ECE2 in neural development is new and interesting. And the use of both human and mouse systems are important. There are concerns, however, on the quality of the data and the interpretation of the results. The migration phenotype is not convincing, and it is not

clear whether it is due to radial glia scaffold change (proteomic data is suggestive of this) or the fate change of the neurons (other experiments) and how the collective phenotypes are relevant for PH.

1. In Figure 1, the apical basal polarity of the cerebral organoids appears to be reversed (1D). In all published data on COs, even with different protocols, the cortical plate is almost always localized to the outside, whereas VZ is inside. In addition, radial glial cells labeled in Fig1G under control conditions showed odd morphology with radial processes stopped underneath the cortical plate, whereas the radial process appeared to be more normal-like under ECE2 KD conditions. This raises significant concerns of the COs generated in this study. In 1E, the expression of ECE2 in apical surface is unclear at all. Is it expressed in radial glia cells or in newly generated neurons? This is an important question as the in utero KD experiments performed in mouse are mostly targeting radial glia cell.

2. In Figure 2, the authors used knockdown approach to investigate the role of ECE2 in the developing mouse brain. As the phenotypes in the mouse brain are mainly related to the radial glia cells, but not neurons, the ECE2 expression in radial glia cells should be examined and the knockdown efficacy needs to be validated properly in these cells. The author should present immunohistochemistry data that showing knockdown of ECE2 in COs and the mouse cortex as well as qRT-PCR data. To exclude potential off-target effect of miRNA, the authors should perform the rescue experiments with overexpression of ECE2.

3. In Fig 3, even though the authors shows that there is no difference in size of organoids between control and ECE2KO, the sample images of COs appeared to be very different between control and KO, more rigorous analysis is required. Again, the phenotypes observed on cell fate is related to radial glia cells, not neurons.

4. In Fig4 F, the quantifications of immunoreactivities are rather arbitrary. Can the authors measure the intensity of immunoreactivity using software such as ImageJ?

5. The data on proteomics are correlative and need to be validated.

6. Throughout the paper, the scale of the control and experiment group (ECE2 KO, KD or PHOS) images are not same (Fig1G, H, Fig 2B, D,E, H, I, Fig3C, F,H, J, L). Typically the example images appear to exaggerate the actual phenotype in the experimental groups. I find this to be very misleading. Also, many images in the figures lack a scale bar.

Referee #3:

Overall, this study uses human brain organoids and in utero electroporation in mouse to investigate the role of a periventricular heterotopia-related gene (ECE2) on cortical development. Specifically, the authors examine cellular phenotypes such as neuronal migration, neuronal differentiation, cytoskeleton structures, macrostructural phenotypes such as nodule-like/rosette-like structures. The candidate ECE2 gene is novel and systematically testing the cellular consequences associated with its loss is important. However, there are several major technical and experimental issues that make the results difficult to interpret. Moreover, the overall conclusions are underwhelming as they describe a number of phenotypes without an overarching model of how the data fits together to explain heterotopia formation.

Major comments:

- The use of RNAi-based knockdown techniques is not strong considering the potential for off-target effects of these techniques and the authors do not use more than one control and one experimental RNAi construct. The overexpression of synthetic RNAi (siRNA or miRNA) expressing plasmids can be problematic due to the extensive number of targets a single miRNA or siRNA can have within a given cell type. Even with in silico predictions to design RNAi with complete homology to a given target mRNA, there can be extensive off targets of potentially hundreds of other mRNAs or disruption of endogenous miRNA processing. Thus, standard practice is to use at minimum 2 RNAi constructs with completely independent sequences targeting the same mRNA. In this manuscript 3 RNAi constructs were tested, but only 1 was eventually used. Moreover, evidence of gene and protein KD should be provided for both mouse and human work.
- The terminology regarding miRNAs is problematic; e.g., "miRNAs ... were injected into the ventricle-like lumen". miRNAs are the end product of primary microRNA (pri-miRNA) processing. Injecting a plasmid expressing a microRNA should be referred to as such. Furthermore, the data regarding the miRNA expression experiment is only understandable after reading the methods; it would be useful to clarify early on that the plasmids used express both RNAi and GFP.
- No details are provided on cell reprogramming and no validation of the hiPSC clones is shown

(pluripotency markers, genome integrity, etc).

- It is great to see the use of isogenic human stem cell lines generated with CRISPR-Cas9. However, characterization of these clones following genome engineering is not included, nor are off-targets investigated.
- It is unclear from the methods section, but it seems the authors don't guide the differentiation of organoids towards a cerebral cortical fate. Therefore, it is likely that cells from multiple brain regions are present as well other non-ectodermal lineages. In fact, the proteomics analysis points to a number of mesenchymal markers being different suggesting that the comparison across conditions is likely not valid. In fact, the paper includes no characterization of neuronal fate of either 2D or 3D differentiation methods. The extended passaging of progenitors could also caudalize neural fate and some of the ventricular zones may not be dorsal forebrain as assumed throughout the paper. This is a critical point as phenotypes in PH are generally restricted to the forebrain. In this context, it is unclear how the extensive quantifications of progenitors have been performed and how they can be interpreted. The use of FACS is also worrisome due to dissociation bias and because these nuclear markers have not been used reliably in flow cytometry experiments.
- The authors identify phenotypes related to disruption of ECE2, but these cellular phenotypes are not clearly related to the development of heterotopias in patients. For example, in the text, it is written that the ectopic localization of neurons after miRNA expression "mimicking the neuroanatomical anomalies observed in patients with PH". This statement is not supported by the data. Defects in neuronal migration of individual cells may be quite different than the clusters of cells within heterotopias in human brains, and not all migration defects result in heterotopias.
- Treatment of cultures with phosphoramidon were shown in this manuscript to cause alterations in migration. Phosphoramidon inhibits other enzymes including metallo-endopeptidases more broadly as well as ECE1. Attributing the phenotype to ECE2 inhibition specifically is not possible using this pharmacological approach.
- As noted in the manuscript, mouse knockouts of ECE2 are available. Can these knockouts at embryonic stages be analyzed to investigate the effect of ECE2 knockout on neuronal migration, or other phenotypes?
- Is figure 2F an inset of figure 2E on the left? The brightness of the GFP signal doesn't appear to be the same. Is the image scaled unequally in the x and y directions?
- Are the data points in Figure 4 G, I, K, M binary? Why are some plots appearing as bars, and others as lines?
- ECE2 does not seem to be expressed in proliferative zones. How is this protein affecting migration? When is it expressed in development? In Figure 3F there seems to be some expression of ECE2 in the KO hiPSC clones. More characterization of those transcripts is needed.
- In several areas claims are overstated and are more appropriate for a discussion section than a results section. PH is often referred to as a 'disorder', but it is likely a group of etiologically heterogeneous disorders.

Minor Comments:

- More details should be provided In Figure legends. For example, what type of gene expression is plotted in Figure 1B?
- NeuN IHC is unconvincing. Expression in human neurons appears quite late in cortical development and these are very early stage experiments.
- When discussing "the morphology of transfected aRG was changed to a less radial and bipolar shape" the figure of relevance is listed as Figure 2A, but this doesn't seem to be relevant to the statement.
- The concluding section is only one sentence and should be extended to emphasize the pertinent findings of this work and their place within the larger context of PH-related studies.

Referee #1:

Isabel *et al.* investigate the role of endothelin-converting-enzyme-2 (ECE2) on a neuronal migration disorder, periventricular heterotopia (PH). Previous studies showed the genetic heterogeneity in PH. The authors examined the role of ECE2 in developing cortex by using mouse model and human cerebral organoids. The authors showed that knockdown (KD) ECE2 in cortical organoids led to the accumulation of neurons in the ventricular zone. Furthermore, the KD of ECE2 on developing mouse cortex resulted in more ectopic rosettes and nodules. Moreover, ECE2 inhibition or knock out resulted in changes in the cytoskeleton and apico-basal polarity. Lastly, the proteomics analysis in organoids deciphered the role of ECE2 in extracellular matrix production and integrity. The manuscript presents multiple comprehensive data to describe the function of ECE2 in developing cortex. Despite the interesting findings, following comments need addressed to improve the manuscript.

1. The authors presented two PH patients containing mutations at *ECE2* and the MRI result. However, the authors examined effects of ECE2 by generating organoids with *ECE2* KD, or KO, as well as KD in mouse brain, based on the hypothesis that the ECE2 mutations found in patients are loss of function. Authors should discuss this more, or generate iPSC or ESC with the given point mutations to investigate the function of the patients' mutation.

Thanks for this important note. The patient mutations were *in silico* predicted to have a damaging effect on the encoded protein, we therefore initially concentrated on KD, KO, inhibition. In addition, several attempts were made to generate IPCs with the patient's mutations and constructs containing the specific variants of *ECE2* using site directed mutagenesis kits, but we were, unfortunately, not successful. Alternatively, we therefore decided to perform, in addition to KD, KO and enzymatic inhibition, a full set of new experiments in which we overexpress *ECE2* both *in vivo* and in cerebral organoids and found that overexpression (OX) of *ECE2* also results in alteration of neuronal positioning, suggesting that critical levels of *ECE2* are essential for proper cortical development (Figures 4 and EV4).

2. In Figure 1C, the expression profile of ECE2 seems an interesting. According the website referenced in the manuscript, some data (ID in the web) for ECE2 show decrease during neural differentiation, and some data show increase during neural differentiation. Authors should describe what each dot represents and provide more representative data. Since the data was not generated by the authors, they may consider move it to supplementary data. Authors may perform qPCR for ECE2, as well as other neuronal, pluripotency, and three germ layer markers during the neural differentiation in 2D and 3D, and present that their neural differentiation was good and expression of ECE2 shows the pattern in Figure 1C.

This comment touches upon a finding on ECE2 that we did not comment on in our manuscript. The common annotation of the *ECE2* gene suggests that the first exons encode a methyltransferase, followed by a transmembrane domain and the luminal metallopeptidase (see UCSC genome browser, older ENSEMBL versions and uniprot). It has been suggested though, based

on expressed sequence tags which do not support this annotation, that the *ECE2* gene locus actually consists of two separate genes (Jakobsson et al., 2017) – one encoding a methyltransferase and one encoding a membrane-metallopeptidase, the correct *ECE2*. The current ENSEMBL release 96 from April 2019 finally incorporated these findings. In the LIBD STEM CELL BROWSER (<http://stemcell.libd.org/scb/>), the former annotation was applied. Thus, exons e267505-8, which as you pointed out show a different decrease in expression with neuronal differentiation, are wrongly annotated and actually encode the methyltransferase eEF1AKMT4, while the real *ECE2* gene starts with e267510 in the LIBD STEM CELL BROWSER. Of course, as the data shown in Figure 1C was not generated by us, we now moved it to the expanded view figures (Figure EVD). Besides, we confirmed cell type identity in our samples by qPCR for NPC and neuronal markers (now in the method section). Another ISH-based database supporting the increased expression of *ECE2* in neurons over NPCs in the neonate brain is the Marmoset gene atlas, which we now cite in the revised version of the manuscript (Shimogori et al., 2018).

3. When analyzing the *in vivo* function of *Ece2* in mouse cortex, sections were made as Bin. Authors should show the defining of each bin.

We added an example of the binning strategy in the Figure EV2B.

4. In Figure 4A, and 4C, it is difficult to see what the F-actin area represents. Figures for control A and C are very different, although they should represent similar apical adherens belt. Authors need to show the diagram of whole organoid, and mark where the given figures represent. Same is true for 4E, and most of the figures given.

The figures represent the area of interest in germinal zones. As every cortical-like region is a little different within organoids of the same condition, it is natural that the control ventricles shown here are not identical. The germinal zones in (former) Fig. 4A (now Figure EV3N) were located inside the COs, whereas those in 4C (now Figure 3H) were facing with the apical surface towards the edge of the COs, which explains that they look different. Here (Figure for Reviewer 1A, Figure EV1B) you can see a overview images of CO sections and additional lower magnification images of ac-tub, F-ACTIN and ARL13B stainings (Figure for Reviewer 1B, C, E).. Since both the main figures and expanded view figures are actually already quite full of information, we think that the specific enlargements contain the most essential info and therefore we would rather not add the overview in the final figures but of course we can.

For (former) Fig 4E, we selected high resolution images of the *en face* structure of the apical F-actin belt and the primary cilia. Here (Figure for Reviewer 1C) we show lower magnification overview images (with V' indicating the position of the ventricle-like lumen) and besides, we clarified the position of the apical surface with a dotted line (Figure 3J) as done for most other images. To further support our findings from ISH on the disruption of the honeycomb structure of the apical belt in *ECE2* KO organoids, we additionally performed fractionation of G- and F-actin by ultracentrifugation of organoid lysate from 4 independent batches plus Westernblot for quantification of both total actin (Figure 3F and Figure EV3L) and the G- to F-actin ratios (Figure

3G and Figure EV3M). This revealed a significant reduction in total actin levels and specifically a reduction in F-actin in *ECE2* KO organoids compared to CTRL.

5. In Figure 4F, G, H, and I, are the acetylated-tubulin staining is showing that all cells in *ECE2* KO and Phos treated organoids showing lower level of acetyl-Tubulin, or only neurons or aRG are showing the lower? It is difficult to see. Authors may perform the staining of Acetyl-Tubulin in the setting of experiments of Figure 1 and Figure 2 where the KD cells vs control can be distinguished by the presence of GFP. Additionally, authors need to perform Acetyl-Tubulin Western blot for quantification.

Acetylated tubulin was reduced throughout the progenitor zone and neuronal layer. Due to the nice parallel organization of stabilized microtubules in the aRG, we selected to present in the figure regions showing the progenitor zone. To prevent possible misunderstanding, this revised version contains images of the full ventricular area (Figure 3K) including VZ-like and CP-like regions in CTRL and *ECE2* KO organoids immunostained for ac-tub. As suggested by the reviewer, we have performed Westernblot on CTRL vs *ECE2* KO COs, but this did not result in a significant change (see Figure for Reviewer 1D). WB analysis is performed on the whole organoid, which contains also non-ventricular area, and therefore the quantification could be compromised. To be more accurate, we performed signal intensity quantification as suggested by the reviewer (see next point and Figure 3L for *ECE2* KO; Figure EV3Q for PHOS treated organoids).

In addition, as for the quantification of the phenotype, authors used 1 or 0, as either high or reduced signal or density. This kind of quantification seems very arbitrary and subjective. authors need to used more objective way to quantify.

We have quantified the signal intensity as suggested (Figure 3L, Figure EV4Q).

6. Figure 4J, and L are areas that supposedly represent the ventricular zone containing aRG, but they do not seem to ventricular zone. Again, authors need to show where the figure represent by using a diagram of organoid. Additionally, authors should present more data showing the similar staining. The so-called representative images of the Figures do not seem to represent the average structures of the organoids.

We added a schematic diagram depicting the selected areas (Figure EV1B). To avoid the concern about biased choice of images for the figure, we are happy to show more examples here (Figure for Reviewer 1E).

7. Western blot for Figure EV4A is not convincing. It is difficult to see the difference in band intensity of CTNNB1. Additionally, authors should present the total level of ERM in Figure EV4D.

The WB shown in Fig EV4D was only serving as an experimental confirmation of the Proteomics data, we therefore decided to remove this panel.

Figure for Reviewer #1: Clarification of image selection and analysis strategy for cytoskeletal and polarity markers. **A**, Overview image of CO section stained for apically localised F-ACTIN belt and primary cilium marker ARL13B using DAPI for nuclear counterstaining shows multiple neuroependymal regions per organoid (Scalebar = 200 μ m). **B**, Further example images of staining for acetylated TUBULIN in CTRL, PHOS-treated and ECE2 KO COs (Scalebar = 100 μ m). **C**, Increase in thickness of the apical F-ACTIN belt upon *ECE2* KO or inhibition in COs. Lower magnification images of F-ACTIN staining combined with DAPI in images shown in Figure 3H and additional images (Scalebar = 100 μ m for CTRL and *ECE2* KO COs, 50 μ m for CTRL and PHOS treated COs). **D**, Example WB for ac-tub in CTRL and *ECE2* KO COs and quantification from n=4 samples of 3 COs each from 4 independent batches (n.s.). **E**, Decrease in apically localised ARL13B upon *ECE2* KO or inhibition in COs. Lower magnification images of ARL13B staining (+DAPI) complementing Figure 3M, N and Figure EV3R, S (Scalebar = 100 μ m). Ventricule-like regions are marked as V'.

Referee #2:

In this manuscript, Buchsbaum *et al* investigated the role of *ECE2*, a potential candidate gene of Periventricular Heterotopia (PH), in neurogenesis during cortical development in mouse and human iPSCs derived cerebral organoids. The main findings of this study are: 1) *ECE2* is identified as a PH candidate gene and enriched in neurons and apical surface in cerebral organoids (COs). 2) *ECE2* knockdown and inhibition of *ECE2* activity cause defects in neuronal migration in COs. 3) *ECE2* knockdown leads to ectopic rosettes and nodules during embryonic cortical development in mouse. 4) Alteration in *ECE2* activity and expression leads to defects in cytoskeleton in COs. 5) Proteome analyses confirmed the role of *ECE2* in cytoskeleton dynamics, and identified its novel role in ECM composition.

These findings on the functional role of *ECE2* in neural development is new and interesting. And the use of both human and mouse systems are important. There are concerns, however, on the quality of the data and the interpretation of the results. The migration phenotype is not convincing, and it is not clear whether it is due to radial glia scaffold change (proteomic data is suggestive of this) or the fate change of the neurons (other experiments) and how the collective phenotypes are relevant for PH.

1. In Figure 1, the apical basal polarity of the cerebral organoids appears to be reversed (1D). In all published data on COs, even with different protocols, the cortical plate is almost always localized to the outside, whereas VZ is inside. In addition, radial glial cells labeled in Fig1G under control conditions showed odd morphology with radial processes stopped underneath the cortical plate, whereas the radial process appeared to be more normal-like under *ECE2* KD conditions. This raises significant concerns of the COs generated in this study. In 1E, the expression of *ECE2* in apical surface is unclear at all. Is it expressed in radial glia cells or in newly generated neurons? This is an important question as the *in utero* KD experiments performed in mouse are mostly targeting radial glia cell.

“Reversed ventricles” with the progenitor zone facing the medium outside the CO appear very commonly in COs generated with the unpatterned approach published by (Lancaster and Knoblich, 2014). Regarding the expression of *ECE2*, as ISH in marmoset brain shows the same pattern of *ECE2* RNA expression (Shimogori et al., 2018), we have no reason to doubt our ISH images.

We understand the concerns about the RG morphology in Figure 1G. The control image in Figure 1G was selected as the size of the electroporated area was similar to that in the KD. We therefore showed additional images for 4 dpe in Figure EV1F. In the revised version of our manuscript, we selected a different image for the control electroporation (new panel in Figure 1E). Concerning the expression of *ECE2* on the protein level (now Figure 1C): *ECE2* is expressed in the apical surface/compartiment (see dotted line) of aRG. Additionally, we found *ECE2* expressed in newly generated neurons co-localizing with MAP2, and also weakly distributed throughout the progenitor zone within radially migrating neurons.

2. In Figure 2, the authors used knockdown approach to investigate the role of ECE2 in the developing mouse brain. As the phenotypes in the mouse brain are mainly related to the radial glia cells, but not neurons, the ECE2 expression in radial glia cells should be examined and the knockdown efficacy needs to be validated properly in these cells. The author should present immunohistochemistry data that showing knockdown of ECE2 in COs and the mouse cortex as well as qRT-PCR data. To exclude potential off-target effect of miRNA, the authors should perform the rescue experiments with overexpression of ECE2.

The knockdown efficiency of the used miRNAs targeting human and mouse *ECE2/Ece2* were tested in human neuroblastoma cells and mouse P19 cells. We added graphs showing the respective qPCR data to the EV Figures (Figure EV1E and EV2A). Further confirmation of micro-RNA KD efficiency in the relevant cell types is not required as it is a common method to confirm KD efficiency in cell lines (Chung et al., 2017). This efficiency should not be different in a different cell type from the same species.

During the revision, we have purchased an overexpression (OX) construct containing ECE2 and performed a full set of new experiments. We have now overexpressed human *ECE2* both in COs and *in vivo*, and performed rescue experiment in *ECE2* KO COs (Figure 4 and EV4). Surprisingly, the neuronal mispositioning (mostly cell non-autonomous) observed upon KD, KO and inhibition of ECE2 is reproduced by OX of *ECE2*. As some of the effects of ECE2 are cell non-autonomous and the results of *ECE2* KD and OX are quite similar we did not combine the *ECE2* OX and *ECE2* micro-RNA as the resulting data could only be confusing as the combination of added phenotypes rather than real rescues. Nevertheless, we performed rescue on *ECE2* KO COs (all cells constitutively lack ECE2) and obtain local rescue of the neuronal mispositioning (Figure 4K, L).

3. In Fig 3, even though the authors show that there is no difference in size of organoids between control and *ECE2* KO, the sample images of COs appeared to be very different between control and KO, more rigorous analysis is required. Again, the phenotype observed on cell fate is related to radial glia cells, not neurons.

As comments of reviewer #2 hint at a need for higher n, we generated an additional batch of CTRL and *ECE2* KO COs and imaged at least 10 COs per cell line every day instead of every 2nd day for the first 2 weeks of culture. This revealed no consistent size difference.

We observed a role for ECE2 both in neural progenitor morphology and directly in neuronal migration. Additionally, we discussed these results in more detail. To get a better impression of the longer-term outcome of cellular composition (cell fate) in COs, we performed FACS analysis at 107 days and found no difference in the number of progenitors, but reduced number of neurons detected also at the early timepoint, suggesting a prominent role of ECE2 in early stages of neurodevelopment (Figure EV3K).

4. In Fig4 F, the quantifications of immunoreactivities are rather arbitrary. Can the authors measure the intensity of immunoreactivity using software such as ImageJ?

Yes, we now measured the signal intensity for acetylated tubulin immunostainings both in the progenitor zone and neuronal layer using ImageJ (Figure 3K, L and EV3 P, Q). Usually, imaging CO sections on the edge of a slide vs in the middle of a slide required adjustment of the laser intensity. But by focusing on sections with comparable position on the object slide, we could circumvent this problem and repeat the analysis in the suggested, more objective way, coming to the same result.

5. The data on proteomics are correlative and need to be validated.

The MS-based proteomic analysis represents state-of-the-art technology for identification of differences in proteomes. In our case comparison of the wt and *ECE2* KO cerebral organoids has clearly identified groups of proteins (laminins, lumican, decorin and collagens) which are part of extracellular matrix and reflects well the observed phenotype from immunohistostaining experiments. Our proteomics data do not stand alone – they overlap with FACS (PAX6, MAP2), and immunohistochemistry (actin, microtubules) results. Taken together, this points towards a more general mechanism underlying the *ECE2* KO effect, which is not carried out by a single protein but rather synergistically by groups of the proteins that we have identified in our proteomics experiment. We used sufficient biological and technical replicates. MS-based proteomics was similarly used on COs by others (McClure-Begley et al., 2018).

6. Throughout the paper, the scale of the control and experiment group (*ECE2* KO, KD or PHOS) images are not same (Fig1G, H, Fig 2B, D, E, H, I, Fig3C, F,H, J, L). Typically the example images appear to exaggerate the actual phenotype in the experimental groups. I find this to be very misleading. Also, many images in the figures lack a scale bar.

We were not aware of this apparent bias and rearranged the figures accordingly and added missing scalebars.

Referee #3:

Overall, this study uses human brain organoids and *in utero* electroporation in mouse to investigate the role of a periventricular heterotopia-related gene (*ECE2*) on cortical development. Specifically, the authors examine cellular phenotypes such as neuronal migration, neuronal differentiation, cytoskeleton structures, macrostructural phenotypes such as nodule-like/rosette-like structures. The candidate *ECE2* gene is novel and systematically testing the cellular consequences associated with its loss is important. However, there are several major technical and experimental issues that make the results difficult to interpret. Moreover, the overall conclusions are underwhelming as they describe a number of phenotypes without an overarching model of how the data fits together to explain heterotopia formation.

In the Schemes in Fig. 5E-F, we have summarized all phenotypes observed upon *ECE2* KD and/or chronic inhibition/KO. Now we added also *ECE2* OX *in vivo* and in COs to our analyses

and summarized the findings in Figure 5G. Our findings suggest a role for ECE2 in multiple pathways known to play a role in cortical development and commonly resulting in neuronal heterotopia upon lack of functional ECE2.

Major comments:

- The use of RNAi-based knockdown techniques is not strong considering the potential for off-target effects of these techniques and the authors do not use more than one control and one experimental RNAi construct. The overexpression of synthetic RNAi (siRNA or miRNA) expressing plasmids can be problematic due to the extensive number of targets a single miRNA or siRNA can have within a given cell type. Even with *in silico* predictions to design RNAi with complete homology to a given target mRNA, there can be extensive off targets of potentially hundreds of other mRNAs or disruption of endogenous miRNA processing. Thus, standard practice is to use at minimum 2 RNAi constructs with completely independent sequences targeting the same mRNA. In this manuscript 3 RNAi constructs were tested, but only 1 was eventually used. Moreover, evidence of gene and protein KD should be provided for both mouse and human work.

We added the results of our qPCR analysis of the micro-RNA efficiency tests for human and mouse constructs to the EV figures (EV1E and EV2A). The shown data for COs was actually generated with 2 different miRNAs and pooled data was presented. We now show the data with the two different constructs which showed the same phenotype (Figure 1F).

- The terminology regarding miRNAs is problematic; e.g., "miRNAs ... were injected into the ventricle-like lumen". miRNAs are the end product of primary microRNA (pri-miRNA) processing. Injecting a plasmid expressing a microRNA should be referred to as such.

This terminology has been previously used in other publications (Cappello et al., 2013; Klaus et al., 2019; O'Neill et al., 2018), but we adjusted the terminology accordingly if the reviewer feels that this nomenclature is inappropriate.

Furthermore, the data regarding the miRNA expression experiment is only understandable after reading the methods; it would be useful to clarify early on that the plasmids used express both RNAi and GFP.

Thanks for pointing out this possibility of misunderstanding. We adjusted the text accordingly.

- No details are provided on cell reprogramming and no validation of the hiPSC clones is shown (pluripotency markers, genome integrity, etc).

We performed 2D and 3D *in vitro* experiments using a published iPSC line (Klaus et al., 2019) in the generation of which standard characterization of pluripotency and karyotyping were performed, which we have now added to the methods and which we are of course ready to share.

- It is great to see the use of isogenic human stem cell lines generated with CRISPR-Cas9. However, characterization of these clones following genome engineering is not included, nor are off-targets investigated.

We added according information to the methods part (same as applied for control line).

- It is unclear from the methods section, but it seems the authors don't guide the differentiation of organoids towards a cerebral cortical fate. Therefore, it is likely that cells from multiple brain regions are present as well other non-ectodermal lineages. In fact, the proteomics analysis points to a number of mesenchymal markers being different suggesting that the comparison across conditions is likely not valid. In fact, the paper includes no characterization of neuronal fate of either 2D or 3D differentiation methods. The extended passaging of progenitors could also caudalize neural fate and some of the ventricular zones may not be dorsal forebrain as assumed throughout the paper. This is a critical point as phenotypes in PH are generally restricted to the forebrain. In this context, it is unclear how the extensive quantifications of progenitors have been performed and how they can be interpreted. The use of FACS is also worrisome due to dissociation bias and because these nuclear markers have not been used reliably in flow cytometry experiments.

As cited in our manuscript, the COs of this study were generated with the unpatterned protocol published by the Knoblich lab (Lancaster and Knoblich, 2014). We are aware of the variability in brain regions generated and therefore performed immunostaining for PAX6 on every germinal zone and in the following only used dorsal cortical regions for further analyses. For FACS, every n consists of 3 combined COs to counteract the variability and avoid misinterpretation of the FACS data and to exclude biased analysis. We have successfully applied and published our former results based on FACS analysis (Klaus et al., 2019) and performed extensive validation experiments beforehand. We are aware that FACS analysis merely gives a first impression of the rough composition of COs. That is why we complemented this analysis with immunostainings and quantification of e.g. of PH3. We also confirmed 2D NPC and neuronal identity by qPCR (methods section). The combination of the different characterization and quantification methods make us confident of the obtained results.

- The authors identify phenotypes related to disruption of ECE2, but these cellular phenotypes are not clearly related to the development of heterotopias in patients. For example, in the text, it is written that the ectopic localization of neurons after miRNA expression "mimicking the neuroanatomical anomalies observed in patients with PH". This statement is not supported by the data. Defects in neuronal migration of individual cells may be quite different than the clusters of cells within heterotopias in human brains, and not all migration defects result in heterotopias.

We rephrased more carefully that. Ectopically positioned neurons within the progenitor zones of CO germinal zones were accepted as resembling PH by reviewers of previous work of our lab (O'Neill et al., 2018) and due to the sparsity of transfected cells, we cannot expect the formation of proper neuronal nodules within a few days as we in contrast recently reported for FAT4 and DCHS1 patient and KO COs (Klaus et al., 2019).

- Treatment of cultures with phosphoramidon were shown in this manuscript to cause alterations in migration. Phosphoramidon inhibits other enzymes including metallo-endopeptidases more broadly as well as ECE1. Attributing the phenotype to ECE2 inhibition specifically is not possible using this pharmacological approach.

We used phosphoramidon at 1 μ M concentration, at which most ECE1 activity is left but ECE2 is inhibited to less than 5 % remaining activity (Emoto and Yanagisawa, 1995). The use of Phosphoramidon was only one of a multitude of methods that we combined in this work, including acute KD and OX in human-derived COs and in the developing mouse cortex and full KO in iPSCs to generate COs from them. All other methods point in the same direction and thus we believe that we can trust the observed phenotype.

- As noted in the manuscript, mouse knockouts of ECE2 are available. Can these knockouts at embryonic stages be analyzed to investigate the effect of ECE2 knockout on neuronal migration, or other phenotypes?

We have contacted the group that generated the mentioned *Ece2* KO mice with exactly that aim – without any reaction. Of course Referee #3 is right that this could be done. We are mostly interested in neuronal migration in human cortical development, though. In the past, other knockout mouse models have failed to recapitulate the PH phenotype seen in human patients, while acute KD has proven to be a more successful method to mimic human neuronal migration phenotypes (Buchsbaum and Cappello, 2019). Therefore, we decided for the fast, efficient, and well-established acute KD (and OX) by *in utero* electroporation.

- Is figure 2F an inset of figure 2E on the left? The brightness of the GFP signal doesn't appear to be the same. Is the image scaled unequally in the x and y directions?

Yes, the image in Figure 2F is a higher resolution of the framed area in 2E. The areas are not 100% the same, but the x-y scaling was of course not changed.

- Are the data points in Figure 4 G, I, K, M binary? Why are some plots appearing as bars, and others as lines?

The single data points for every n were binary, yes. We double-checked the graphs, thanks for this note.

We have now quantified the acetylated tubulin signal in Fiji as the mean grey value and show the results as z-scores (new Figure 3L, Figure EV3Q), while the ARL13B data (now Figure 3N and Figure EV3S) is binary. In the box plots, the mean is shown as a red line and the median as black line. Boxes represent 25th and 75th percentiles and whiskers extend to 10th and 90th percentiles. So depending on the data distribution, it may appear as bars or as lines with or without whiskers.

- ECE2 does not seem to be expressed in proliferative zones. How is this protein affecting migration? When is it expressed in development? In Figure 3F there seems to be some expression of ECE2 in the KO hiPSC clones. More characterization of those transcripts is needed.

qPCR analysis and IHC showed a low level of *ECE2* expression in NPCs/in the progenitor zone (now Figure EV1C and Figure 1B, C) with a focused concentration of the protein to the apical surface in the progenitors (Fig 1C). Low levels of this peptidase can very well influence migration. Products of *ECE2* activity, such as identified ECM components, can further be produced in and secreted by neurons and from the CP distribute in the extracellular space.

The low leftover level of *ECE2* RNA in the KO iPSC clones (now Figure EV3D) can only stem from DNA contamination of the extracted RNA from CTRL and *ECE2* KO iPSCs in which we did not use DNase. To further characterize the KO lines and confirm absence of functional *ECE2*, we performed ISH on *ECE2* COs (Figure 3B).

- In several areas claims are overstated and are more appropriate for a discussion section than a results section. PH is often referred to as a 'disorder', but it is likely a group of etiologically heterogenous disorders.

We changed the text in the revised version.

Minor Comments:

- More details should be provided In Figure legends. For example, what type of gene expression is plotted in Figure 1B?

Figure 1B (now Figure EV1C) shows qPCR results of *ECE2* expression in NPCs and neurons after normalization to GAPDH. We provided more details in the legends and in the methods section although the journal guidelines were already overshot.

- NeuN IHC is unconvincing. Expression in human neurons appears quite late in cortical development and these are very early stage experiments.

From fetal tissue we know that NEUN (*RBFOX3*) appears rather early in cerebral organoids and human developing brain (Klaus et al., 2019; Nowakowski et al., 2017).

- When discussing "the morphology of transfected aRG was changed to a less radial and bipolar shape" the figure of relevance is listed as Figure 2A, but this doesn't seem to be relevant to the statement.

Thanks for drawing our attention to this mistake – the correct image numbers are now Figure 2E, F and Figure EV2 D, F.

- The concluding section is only one sentence and should be extended to emphasize the pertinent findings of this work and their place within the larger context of PH-related studies.

We extended our summary section as requested but since we are aiming at a report, we tried to keep it short.

References:

- Buchsbaum, I. Y. and Cappello, S.** (2019). Neuronal migration in the CNS during development and disease: insights from in vivo and in vitro models. *Development* **146**,.
- Cappello, S., Gray, M. J., Badouel, C., Lange, S., Einsiedler, M., Srour, M., Chitayat, D., Hamdan, F. F., Jenkins, Z. A. and Morgan, T.** (2013). Mutations in genes encoding the cadherin receptor-ligand pair DCHS1 and FAT4 disrupt cerebral cortical development. *Nat. Genet.* **45**, 1300–1308.
- Chung, H.-C., Nguyen, V.-G., Oh, W.-T., Huynh, T.-M.-L., Moon, H.-J., Lee, J.-H., Kim, H.-K., Park, S.-J. and Park, B.-K.** (2017). Inhibition of Porcine Endogenous Retrovirus by Multi-Targeting Micro RNA Against Long Terminal Region. *Transplant. Proc.* **49**, 2225–2232.
- Emoto, N. and Yanagisawa, M.** (1995). Endothelin-converting enzyme-2 is a membrane-bound, phosphoramidon-sensitive metalloprotease with acidic pH optimum. *J. Biol. Chem.* **270**, 15262–8.
- Jakobsson, M. E., Malecki, J., Nilges, B. S., Moen, A., Leidel, S. A. and Falnes, P. Ø.** (2017). Methylation of human eukaryotic elongation factor alpha (eEF1A) by a member of a novel protein lysine methyltransferase family modulates mRNA translation. *Nucleic Acids Res.* **45**, 8239–8254.
- Klaus, J., Kanton, S., Kyrousi, C., Ayo-Martin, A. C., Di Giaimo, R., Riesenberger, S., O'Neill, A. C., Camp, J. G., Tocco, C., Santel, M., et al.** (2019). Altered neuronal migratory trajectories in human cerebral organoids derived from individuals with neuronal heterotopia. *Nat. Med.* **1**.
- Lancaster, M. A. and Knoblich, J. A.** (2014). Generation of cerebral organoids from human pluripotent stem cells. *Nat Protoc* **9**, 2329–2340.
- McClure-Begley, T. D., Ebmeier, C. C., Ball, K. E., Jacobsen, J. R., Kogut, I., Bilousova, G., Klymkowsky, M. K. and Old, W. M.** (2018). Cerebral organoid proteomics reveals signatures of dysregulated cortical development associated with human trisomy 21. *bioRxiv* 315317.
- Nowakowski, T. J., Bhaduri, A., Pollen, A. A., Alvarado, B., Mostajo-Radji, M. A., Di Lullo, E., Haeussler, M., Sandoval-Espinosa, C., Liu, S. J., Velmeshev, D., et al.** (2017). Spatiotemporal gene expression trajectories reveal developmental hierarchies of the human cortex. *Science* **358**, 1318–1323.
- O'Neill, A. C., Kyrousi, C., Einsiedler, M., Burtscher, I., Drukker, M., Markie, D. M., Kirk, E. P., Götz, M., Robertson, S. P. and Cappello, S.** (2018). Mob2 Insufficiency Disrupts Neuronal Migration in the Developing Cortex. *Front. Cell. Neurosci.* **12**, 57.
- Shimogori, T., Abe, A., Go, Y., Hashikawa, T., Kishi, N., Kikuchi, S. S., Kita, Y., Niimi, K., Nishibe, H., Okuno, M., et al.** (2018). Digital gene atlas of neonate common marmoset brain. *Neurosci. Res.* **128**, 1–13.

Thank you for your patience while your revised manuscript was assessed by the referees. We have now received the comments as well as cross-comments, and all are pasted below.

As you will see, while referees 1 and 2 are more positive, referee 3 remains unconvinced that the current set of data is sufficiently strong to support the main conclusions. However, referees 1 and 2, while agreeing that the points raised by referee 3 are correct, disagree with her/his final assessment, and indicate that the manuscript can be published if all remaining concerns can be successfully addressed, along the lines suggested by the referees. I would therefore like to invite you to revise your manuscript a second time. Regarding the name of the antisense RNAs used, if the supplier calls them miRNAs, this is fine, but it should be explained in the main manuscript text when they are first mentioned.

A few other changes will also be required:

The manuscript is laid out as a scientific report with combined results and discussion sections and 5 main figures, but the character count largely exceeds the report limit (45.000 characters instead of 29.000). You can either reduce the character count to below 35.000 (counting characters with spaces EXCLUDING materials and methods and references but including the main figure legends), or you add one more main figure to have at least 6 main figures and separate the results and the discussion sections to publish your study as a full article. You can find more information in our guide to authors online.

The Data Availability Section is incomplete, it needs to list the accession ID and the direct link to the website where the data is deposited. Please correct.

I attach to this email a manuscript word file with comments by our data editors. Please address all comments using the track changes option and upload a revised word file with your final manuscript submission.

Please add up to 5 keywords.

Please rename the Conflict of Interest statement to exactly that.

The manuscript text is missing a callout to Fig 5F and to the movies, these need to be added.

The EV movie legends need to be zipped with their respective movie files, and for each movie a zipped file needs to be uploaded.

The EV figure legends need to be added to the main manuscript file to after the main figure legends.

The EV table file contains 9 EV tables, these must be separated and uploaded as individual files. If some of the files belong together, they can also be uploaded together. Please note that all EV tables must be cited somewhere in the manuscript file.

The manuscript text calls out Table S3 and S4, but no such tables exist.

Please move the information about the funders to the acknowledgement section.

Please delete the word "table" from the table with primary antibodies. Everything that is called a Table needs to be numbered and cited. It is fine to list the antibodies and miRNA sequences in a table format in the method section without a citation.

EMBO press papers are accompanied online by A) a short (1-2 sentences) summary of the findings and their significance, B) 2-3 bullet points highlighting key results and C) a synopsis image that is 550x200-400 pixels large (the height is variable). You can either show a model or key data in the

synopsis image. Please note that text needs to be readable at the final size. Please send us this information along with the revised manuscript.

With my best wishes for a merry Christmas and a happy, healthy and successful new year.

REFEREE REPORTS

Referee #1:

Isabel et al. revised the manuscript according to the comments. There are some concerns listed below.

1. As pointed out in the original version of manuscript, it is essential that what is the orientation of the organoids and authors give the diagram. Authors also acknowledged that they used un-guided method by Lancaster, which produce organoids that show the upside-down ventricular zone. Because the proliferation, migration, and differentiation of neuroprogenitors at ventricular zone are most likely dependent on the environmental factors (e.g. paracrine, or physical interaction), it is even more important that authors distinguish the effect of where ventricular zone is located.
2. Authors claim that the electroporation of plasmid in organoids deliver the DNA into ventricular zone radial glia cells. However, there is no data supporting this claim. In addition, along the same line as #1 above, if the VZ is exposed to outside, there may be areas in organoids that GFP+ plasmids are electroporated into neurons. In terms of the defining the VZ throughout the manuscript, it is arbitrary in defining VZ. PAX6+ staining should define the VZ better than the absence of NeuN. Only murine cortex staining shows the PAX6+ area as VZ, but when characterizing the human brain organoids, few PAX6 staining is used.
3. The conclusion of the paper is really confusing. In Figure 4, authors present the data that overexpression of ECE2 showed similar phenotypes as depletion of ECE2. Authors may discuss further on the issue. Authors should present models how the contradictory findings make sense in the role of ECE2 regulating differentiation, migration, or other aspect of RG.
4. In Figure 5E, what each figure represents is not clear? Authors should give description.

Referee #2:

In general, the authors have done a good job in revising their manuscript and responding to most of my concerns. However, the reviewers failed to address one of my comments regarding the scale of representative images. In their rebuttal letter the authors claim to have rearranged the figures, but I do not see the changes in some figures (Fig2D, H, and I, and Fig4D, G and K). I strongly suggest to replace the images so the scales of images for control and experimental group are same. When this revision is made, I support the publication of this manuscript.

Referee #3:

In this revision, the authors attempt to address the technical and experimental issues raised in the first review. I was glad to see improvements in the phrasing, like for instance changing "mimicking the neuroanatomical anomalies observed in patients with PH" to "mimicking some of the neuronal migration defects, e.g. ectopic neurons, observed in patients with PH". It is concerning that 2 different RNAi constructs were previously pooled in the initial submission's figures, although Figure 1F is improved now.

With regards to phosphoramidon treatment experiments, the authors claim that "After inhibition of ECE2 activity, neurons migrated significantly more slowly, rested more often and migrated in a more tortuous path compared to control neurons. Thus, the activity of ECE2 is required for normal neuronal dynamics, suggesting an additional cell-autonomous function of ECE2 directly in neuronal

migration." Was the activity of ECE2 tested after phosphoramidon treatment? This experiment may support prior findings but is not sufficient to make the above claims. Additionally, how does this experiment support determination of cell autonomous activity, since the compound is bath applied to all cells?

Use of shRNAs is well established to cause neuronal migration defects, irrespective of sequence (PMID: 24945770). More experimental investigation is necessary to establish that the phenotypes demonstrated are a result of ECE2 knockdown. This is a key issue as this is one of major experiments/claims in the paper. Currently there is a lack of evidence for knockdown of ECE2 on the RNA or protein level with the shRNAs used. Moreover, with only two shRNAs and one control shRNA (unclear if an shRNA was used in the control, or whether it was a mock electroporation) used, it is unconvincing whether this is an effect of specific ECE2 knockdown.

The KD experiments in mouse cortex or COs was also done with RNAi experiments, but the data is not quantified with respect to the particular RNAi construct used. It is unclear if one or two were used, which ones were used, or if they were pooled. There is no mention of specific shRNAs used in the entirety of Fig. 2. Data from individual shRNAs should not be pooled.

In one section, it is claimed that KD will be performed acutely in vivo and vitro. However, the section is unrelated to the acute KD in vivo or in vitro.

Regarding miRNA terminology, the authors rebut that their initial terminology was used in previous publications. All three papers they reference in this regard were from their own group.

The way the authors refer to the constructs microRNAs is potentially problematic- are these constructs functioning as miRNAs? This would mean they target up to hundreds of mRNA targets (PMID: 26372022). It appears that the authors are intending for their use as shRNAs, not miRNAs. This should be clarified.

Cross-comments by referee 1 (in response to author reply):

1. As for the diagram of ventricular zone, it is simple to show where the ventricular zone is located in the whole organoid. It should be easy to add a drawing beside of the picture where the given picture was taken in the organoid.
2. If authors have performed PAX6 staining in human organoids, they should present them as main figures or even as supplementary figures. Why not?
3. 4. Fine.

And in response to referee 3's cross-comments:

I am OK with publishing the paper.

One of major troubles in the paper was that authors added several data without good flow of science, and it was very difficult to read and understand the paper.

As for the technical issues, I agree with referee #3's concern about the use of phosphoramidon and miRNA. But, the issue does not seem that detrimental.

Cross-comments by referee 2:

Regarding the migration assay with phosphoramidon treatment, the migration of differentiated neurons reflects cell-autonomous effect of ECE2 inhibition, which does not rely on the other cell types (as in Fig1G). Although the authors did not investigate the non cell-autonomous function of ECE2 in the neuronal migration further, the data seemed to be sufficient to make the authors' claim. The authors should try to explain the assay more clearly in the method section.

Regarding the knockdown experiment, the referee 3 raised the valid point in terms of terminology

and clarification throughout the manuscript. Clearly miRNA is a wrong term, and the authors should make a correction to either RNAi or shRNA. Also, the authors should clarify which shRNA they used for their experiments. I do see the validation of knockdown efficiency of two shRNAs using RT-PCR in Fig EV1E. And the major phenotype was confirmed with secondary shRNA in figure 1. So, a minor revision should be sufficient.

Cross-comments by referee 3:

I believe one of the other reviewer has also expressed strong concerns. And again, just to make it clear, I believe that the conclusions are not supported by the data because of several technical issues including the use of plasmid-based knockdown, lack of validation for the miRNA targeting constructs to make sure they aren't having off target effects, and the weak pharmacological manipulation with drugs that are not specific.

2nd Revision - authors' response

18 February 2020

Referee 1:

Isabel et al. revised the manuscript according to the comments. There are some concerns listed below.

1. As pointed out in the original version of manuscript, it is essential that what is the orientation of the organoids and authors give the diagram. Authors also acknowledged that they used un-guided method by Lancaster, which produce organoids that show the upside-down ventricular zone. Because the proliferation, migration, and differentiation of neuroprogenitors at ventricular zone are most likely dependent on the environmental factors (e.g. paracrine, or physical interaction), it is even more important that authors distinguish the effect of where ventricular zone is located. *As our patients have broad defects in their brains, we initially decided to use the protocol developed by Lancaster to have a wider view on the role of ECE2 in the developing brain. This protocol, since undirected, has on one side the disadvantage of being more variable in terms of similarity of ventricles but had the advantage of being a bit more complex in terms of cellular composition. We analyze these types of organoids by quantifying many differentiation batches and several different ventricles for each organoid. We do not exclude ventricles to be unbiased. To complement standard quantification, we implemented our analysis and included FACS analysis, precisely to avoid biased quantifications and selection of organoids and ventricles.*

2. Authors claim that the electroporation of plasmid in organoids deliver the DNA into ventricular zone radial glia cells. However, there is no data supporting this claim. In addition, along the same line as #1 above, if the VZ is exposed to outside, there may be areas in organoids that GFP+ plasmids are electroporated into neurons. *Electroporation in organoids is performed in the same way as in the developing mouse brain. We cannot exclude the possibility that a few neurons may be electroporated but since we inject plasmids in the ventricular lumen, the plasmids should in theory only impact radial glial cells, similarly to the electroporation in the mouse brain. Additionally, we exclude all the ventricles that are not electroporated in the correct orientation.*

In terms of the defining the VZ throughout the manuscript, it is arbitrary in defining VZ. PAX6+ staining should define the VZ better than the absence of NeuN. Only murine cortex staining shows the PAX6+ area as VZ, but when characterizing the human brain organoids, few PAX6 staining is used. *We always perform Pax6 staining to confirm dorsal identity. As Pax6+ ventricular zones are at this stage always complementary to Map2+ or NeuN+ area, we show in the figures only the most relevant immunostainings.*

3. The conclusion of the paper is really confusing. In Figure 4, authors present the data that overexpression of ECE2 showed similar phenotypes as depletion of ECE2. Authors may discuss further on the issue. Authors should present models how the contradictory findings make sense in the role of ECE2 regulating differentiation, migration, or other aspect of RG.

We can certainly expand the discussion on ECE2, we were also initially surprised to observe similar effects upon KD and OX of ECE2, but especially considering the changes that ECE2 promotes in the niche, particularly the extracellular matrix, we can suggest that local modifications can disrupt the normal signal distribution and impact on migration and differentiation.

4. In Figure 5E, what each figure represents is not clear? Authors should give description.

We tried to summarize the phenotypes that we observed, we can add more description or remove the scheme, if unnecessary.

Referee 2:

In general, the authors have done a good job in revising their manuscript and responding to most of my concerns. However, the reviewers failed to address one of my comments regarding the scale of representative images. In their rebuttal letter the authors claim to have rearranged the figures, but I do not see the changes in some figures (Fig2D, H, and I, and Fig4D, G and K). I strongly suggest to replace the images so the scales of images for control and experimental group are same. When this revision is made, I support the publication of this manuscript.

We can change the images and adjust the scales, according to the request.

Referee 3:

In this revision, the authors attempt to address the technical and experimental issues raised in the first review. I was glad to see improvements in the phrasing, like for instance changing "mimicking the neuroanatomical anomalies observed in patients with PH" to "mimicking some of the neuronal migration defects, e.g. ectopic neurons, observed in patients with PH". It is concerning that 2 different RNAi constructs were previously pooled in the initial submission's figures, although Figure 1F is improved now.

We apologize for initially pooling the data from the 2 miRNAs, as we observed very similar effects we thought that this could simplify the data and not be a major problem.

With regards to phosphoramidon treatment experiments, the authors claim that "After inhibition of ECE2 activity, neurons migrated significantly more slowly, rested more often and migrated in a more tortuous path compared to control neurons. Thus, the activity of ECE2 is required for normal neuronal dynamics, suggesting an additional cell-autonomous function of ECE2 directly in neuronal migration." Was the activity of ECE2 tested after phosphoramidon treatment? This experiment may support prior findings but is not sufficient to make the above claims.

We agree with the reviewer, we have tried to test the ECE2 enzymatic activity but also had huge variability in results (even of the same samples) which suggests that the available activity tests are not reliable, at least in our system. For this reason, we used phosphoramidon at 1 μ M concentration, as previously published results show that at this concentration most ECE1 activity is left but ECE2 is inhibited to less than 5 % remaining activity (Emoto and Yanagisawa, 1995). Most importantly, the data were confirmed genetically by KD and KO and therefore we believe that the similar outcome supports the phosphoramidon results. This is just one of the multiple methods that we use to show the function of ECE2.

Additionally, how does this experiment support determination of cell autonomous activity, since the compound is bath applied to all cells?

This specific analysis is the only one that we performed in 2D culture, where neurons and NPCs are sparse and not in contact, suggesting this specific phenotype to be a cell-autonomous mechanism.

Use of shRNAs is well established to cause neuronal migration defects, irrespective of sequence (PMID: 24945770). More experimental investigation is necessary to establish that the phenotypes demonstrated are a result of ECE2 knockdown. This is a key issue as this is one of major experiments/claims in the paper. Currently there is a lack of evidence for knockdown of ECE2 on the RNA or protein level with the shRNAs used. Moreover, with only two shRNAs and one control shRNA (unclear if an shRNA was used in the control, or whether it was a mock electroporation) used, it is unconvincing whether this is an effect of specific ECE2 knockdown.

The KD experiments in mouse cortex or COs was also done with RNAi experiments, but the data in not quantified with respect to the particular RNAi construct used. It is unclear if one or two were

used, which ones were used, or if they were pooled. There is no mention of specific shRNAs used in the entirety of Fig. 2. Data from individual shRNAs should not be pooled.

Unfortunately, we failed in being clear here and have generated some misunderstandings. To clarify, we use miRNAs, the nomenclature used by the paper cited by the reviewer (PMID: 24945770) and not shRNAs. In this specific paper cited by the reviewer (PMID: 24945770) it has been shown that shRNAs BUT NOT miRNAs may have an effect on neuronal migration. We did not use shRNAs.

In addition, all miRNAs were tested and validated (Fig. EV1E, EV2A).

For the mouse experiments we have tested 3 miRNAs and 2 of them had unfortunately low effect on expression levels of Ece2 and therefore we performed experiments with the one with highest efficiency.

In one section, it is claimed that KD will be performed acutely in vivo and vitro. However, the section is unrelated to the acute KD in vivo or in vitro.

We can adjust the mistake of course.

Regarding miRNA terminology, the authors rebut that their initial terminology was used in previous publications. All three papers they reference in this regard were from their own group.

The paper cited by the reviewer (PMID: 24945770) uses our same terminology.

The way the authors refer to the constructs microRNAs is potentially problematic- are these constructs functioning as miRNAs? This would mean they target up to hundreds of mRNA targets (PMID: 26372022). It appears that the authors are intending for their use as shRNAs, not miRNAs. This should be clarified.

Micro-RNAs (miRNAs) targeting human or mouse ECE2 were generated using the BLOCK-iT system from Invitrogen (Thermo Fisher). Micro-RNA sequences were determined using Invitrogen's RNAi design tool

<https://rnaidesigner.thermofisher.com/rnaiexpress/setOption.do?designOption=mirnapid=1961720787891316464>.

The terminology was changed in the revised version according to the reviewer's request. We are happy to stay with the original terminology, according to the Invitrogen's design tool: miRNAs.

Accepted

21 February 2020

I am very pleased to accept your manuscript for publication in the next available issue of EMBO reports. Thank you for your contribution to our journal.

YOU MUST COMPLETE ALL CELLS WITH A PINK BACKGROUND ↓
PLEASE NOTE THAT THIS CHECKLIST WILL BE PUBLISHED ALONGSIDE YOUR PAPER

Corresponding Author Name: CAPPELLO
Journal Submitted to: EMBO REPORTS
Manuscript Number: EMBOR-2019-48204V2